# Recursive Bayesian Networks: Generalising and Unifying Probabilistic Context-Free Grammars and Dynamic Bayesian Networks

**Robert Lieck**[*]
Digital and Cognitive Musicology Lab
École Polytechnique Fédérale de Lausanne
1015 Lausanne, Switzerland
`research@robert-lieck.com`

**Martin Rohrmeier**
Digital and Cognitive Musicology Lab
École Polytechnique Fédérale de Lausanne
1015 Lausanne, Switzerland
`martin.rohrmeier@epfl.ch`

## Abstract

Probabilistic context-free grammars (PCFGs) and dynamic Bayesian networks (DBNs) are widely used sequence models with complementary strengths and limitations. While PCFGs allow for nested hierarchical dependencies (tree structures), their latent variables (non-terminal symbols) have to be discrete. In contrast, DBNs allow for continuous latent variables, but the dependencies are strictly sequential (chain structure). Therefore, neither can be applied if the latent variables are assumed to be continuous and *also* to have a nested hierarchical dependency structure. In this paper, we present Recursive Bayesian Networks (RBNs), which generalise and unify PCFGs and DBNs, combining their strengths and containing both as special cases. RBNs define a joint distribution over tree-structured Bayesian networks with discrete or continuous latent variables. The main challenge lies in performing *joint* inference over the exponential number of possible structures and the continuous variables. We provide two solutions: 1) For arbitrary RBNs, we generalise inside and outside probabilities from PCFGs to the mixed discrete-continuous case, which allows for maximum posterior estimates of the continuous latent variables via gradient descent, while marginalising over network structures. 2) For Gaussian RBNs, we additionally derive an analytic approximation of the marginal data likelihood (evidence) and marginal posterior distribution, allowing for robust parameter optimisation and Bayesian inference. The capacity and diverse applications of RBNs are illustrated on two examples: In a quantitative evaluation on synthetic data, we demonstrate and discuss the advantage of RBNs for segmentation and tree induction from noisy sequences, compared to change point detection and hierarchical clustering. In an application to musical data, we approach the unsolved problem of hierarchical music analysis from the raw note level and compare our results to expert annotations.

## 1   Introduction

Long-term dependencies with a nested hierarchical structure are one of the major challenges in modelling sequential data. This type of dependencies is common in many domains, such as natural

---

[*]corresponding author, code at `https://github.com/robert-lieck/RBN`

35th Conference on Neural Information Processing Systems (NeurIPS 2021).

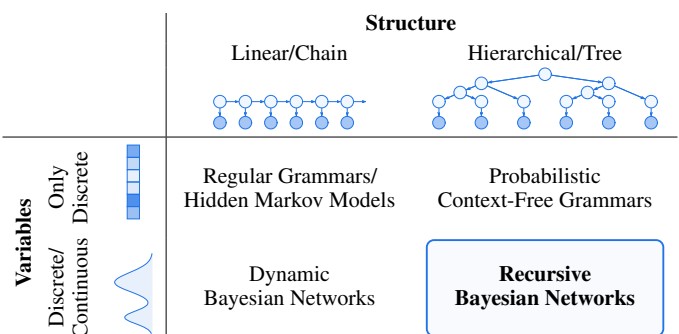

Figure 1: RBNs generalise PCFGs by allowing for continuous latent variables and DBNs by incorporating nested hierarchical dependencies.

language [1], music [2, 3], or decision making [4, 5]. Two of the most widely used probabilistic models for sequential data are probabilistic context-free grammars (PCFGs) and dynamic Bayesian networks (DBNs), both having complementary strengths.

PCFGs are well-established and widely used for modelling hierarchical long-term dependencies in symbolic data [1, 6, 7, 8, 9, 5, 10]. They generalise local (Markov) transition models by allowing for infinitely many levels of nested hierarchical dependencies and a flexible number of latent variables. However, parsing methods such as the Cocke-Younger-Kasami (CYK) algorithm [11, 12, 13, 14, 6] rely on the discrete nature of the rules and variables.

In contrast, DBNs are sequential models with a fixed set of random variables that reoccur at each time step [15, 16]. The variables at each time step may be discrete or continuous, latent or observed, and may have an arbitrary non-cyclic dependency structure among each other, with additional links from the previous and to the next time slice. They comprise important model classes as special cases, such as hidden Markov models (HMMs) if there is only a single discrete latent variable or linear dynamical systems if all dependencies are linear Gaussians [17, 18, 16]. However, DBNs only allow for a fixed chain of Markov dependencies between time slices and cannot represent nested hierarchical structures.

In this paper, we present Recursive Bayesian Networks (RBNs), a novel class of probabilistic models that combines the strengths of PCFGs and DBNs by allowing for nested hierarchical dependencies in combination with arbitrary discrete or continuous random variables (Figure 1). Our main contributions are as follows:

1. With RBNs, we provide a unified theoretical framework for a large class of important sequence models, including PCFGs and DBNs.
2. We generalise inside and outside probabilities from PCFGs to continuous latent variables, allowing for maximum posterior (MAP) inference in arbitrary RBNs.
3. For Gaussian RBNs, we derive an analytic approximation for the marginal likelihood and marginal posterior distribution, allowing for robust parameter optimisation and Bayesian inference.
4. We provide a quantitative evaluation on synthetic data and an application to the challenging task of hierarchical music analysis.

## 1.1 Related Work

PCFGs have a long tradition for modelling nested hierarchical dependencies in symbolic data with a variety of parsing algorithms for inferring the structure and variables' values [13, 6, 14]. Beyond their application to sequential data, PCFGs have been generalised to graph structures [19, 20, 21, 22], which readily transfers to applications of RBNs. Latent vector grammars (LVeGs) [23] are an extension of latent variable grammars (LVGs) [24, 25] with continuous latent states. As for RBNs, approximate parsing is possible in the Gaussian case. However, both LVGs and LVeGs are special cases of RBNs and do not draw the connection to graphical models. More recently, the availability of automatic differentiation libraries, such as `PyTorch` [26], has lead to a number of applications where gradients are propagated through the entire parsing process [27, 28, 29, 30].

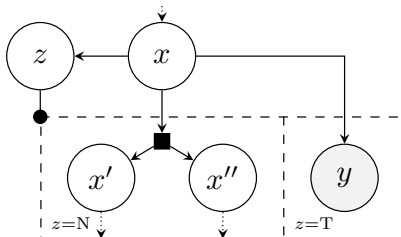

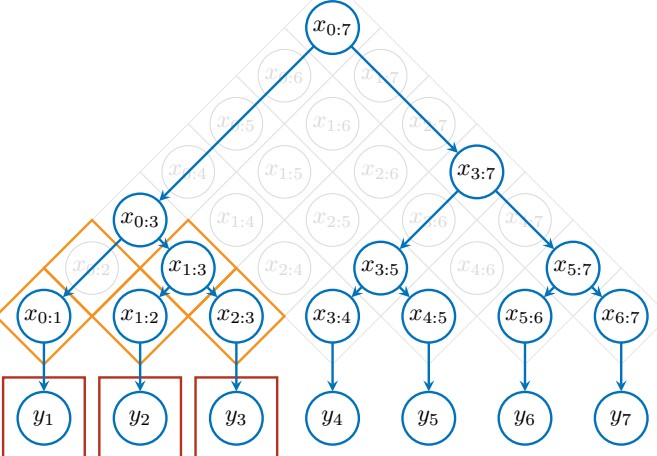

Figure 2: RBN in Chomsky normal form. The non-terminal transition $p_N(x', x'' \mid x)$ and terminal transition $p_T(y \mid x)$ are grouped into an RBN cell with a structural distribution $p_S(z \mid x)$. We use gates [58] to describe structural distributions and extended factor graph notation [black squares; 59] for conditional joint distributions. Considering all possible ways how $n$ observations can be generated by recursively applying the RBN cell produces an RBN chart, as shown in Figure 3.

Figure 3: Chart of an RBN in CNF for sequential data of length $n = 7$. The network obtained by fixing one specific dependency structure is highlighted in blue; latent non-terminal variables that are not part of this particular structure are shown in grey. Orange and red boxes, respectively, indicate the subsets $\mathbf{X}_{0:3}$ and $\mathbf{Y}_{0:3}$ of latent non-terminal and observed terminal variables generated from $x_{0:3}$.

The process of parsing a PCFG or RBN can be formally rewritten as a sum-product network [SPN; 31, 32, 33]. Factor graph grammars [FGGs; 34] generalise PCFGs, case-factor diagrams [35] and SPNs by using a hyperedge replacement graph grammar [19] to describe a distribution over graph structures that is more general than that of RBNs (not only trees). However, none of the approaches addresses the problem of inference with continuous variables that we are facing in RBNs (exponentially many terms with exponentially many nested integrals).

A wide range of probabilistic and neural models operate with a fixed graphical structure and are loosely related to RBNs. Hidden tree Markov models [36, 37] generalise HMMs from chain to fixed tree structures. They model data at each node as observations of a latent Markov process on the underlying tree, which is part of the input data. Additionally estimating the underlying tree structure has been addressed in [38, 39, 40]. Recursive neural tensor networks [41] use a PCFG for parsing a given sequence of symbols to obtain a tree structure, which is then fixed and used as the backbone for a neural network. More generally, there is a number of methods for inferring a fixed structure for graphical models [42, 16, 43, 44], SPNs [45, 46, 47, 48, 32, 49], or graph neural networks [50, 51, 52]. All these methods have in common that a fixed structure is either given or estimated but not treated in a probabilistic Bayesian manner.

Some approaches attempt a Bayesian treatment of the unknown structure of a graphical model or SPN via dynamic programming [53, 54] or Markov chain Monte-Carlo sampling [55, 56, 57]. However, the structure is assumed to be independent of the latent variables (they only become dependent *conditional* on the data) and the latent variables cannot be used to *control* the structure, as it is the case in RBNs. The challenge of continuous variables also remains unsolved.

## 2 Recursive Bayesian Networks

RBNs are *template-based* graphical models that define a joint distribution over network structures and variables' values. The number of template variables is fixed, but the number of instantiated variables, their connectivity and values are governed by the joint distribution. As a rough analogy, RBNs can be thought of as DBNs that can not only be connected linearly to form a chain but also hierarchically to form a tree structure. Alternatively, they can be thought of as a PCFG in which each symbol is a (possibly continuous) random variable.

## 2.1 Definition

RBNs have three types of template variables: 1) latent non-terminal variables (discrete or continuous), 2) observed terminal variables (discrete or continuous), and 3) latent structural variables (always discrete). In the simplest case, illustrated in Figure 2, an RBN has one template variable of each type. Formally, an RBN is defined as follows:

**Definition 1** (Recursive Bayesian Network). *An RBN is a tuple* $(\mathcal{X}, \mathcal{Y}, \mathcal{Z}, \mathcal{T}, \mathcal{S}, p_\mathrm{P})$ *with*

$$\mathcal{X}: \text{a set of latent non-terminal template variables} \tag{1}$$

$$\mathcal{Y}: \text{a set of observed terminal template variables} \tag{2}$$

$$\mathcal{Z}: \text{a set of latent structural template variables, paired up with the non-terminal variables} \tag{3}$$

$$\mathcal{T}: \text{a set of transition distributions } p(v_1, \ldots, v_\eta \mid x) \text{ from a single non-terminal variable } x \in \mathcal{X}$$
$$\text{to a set of non-terminal and/or terminal variables } v_1, \ldots, v_\eta \in \mathcal{X} \cup \mathcal{Y} \tag{4}$$

$$\mathcal{S}: \text{a set of structural distributions } p(z \mid x), \text{ one for each non-terminal/structural pair} \tag{5}$$

$$p_\mathrm{P}: \text{a prior/start distribution for exactly one non-terminal variable.} \tag{6}$$

*The cardinality of a structural variable* $z \in \mathcal{Z}$ *corresponds to the number of possible transitions from the associated non-terminal variable* $x \in \mathcal{X}$; $\eta$ *in* (4) *is called the arity of the transition.*

Generating with an RBN is straightforward. We start by sampling the value of the first non-terminal variable $x$ from the prior distribution $p_\mathrm{P}(x)$ and then repeat the following steps until no unprocessed non-terminal variables are left:

1. sample the value of the associated structural variable from $p(z \mid x)$
2. choose a transition distribution $p(v_1, \ldots, v_\eta \mid x)$ based on the structural variable's value
3. sample the variables $v_1, \ldots, v_\eta$ from the transition distribution
4. for all newly generated non-terminal variables, go to step 1.

The major challenge and focus of this paper is to perform joint inference over the latent structure and non-terminal variables' values conditional on a given set of observations.

**Chomsky Normal Form:** In the simplest non-trivial case, an RBN has one latent non-terminal, one observed terminal, and one latent structural template variable, with one non-terminal transition of arity $\eta = 2$ and one terminal transition of arity $\eta = 1$, as illustrated in Figures 2 and 3. It is defined by four distributions

$$p_\mathrm{P}(x): \text{prior/start distribution} \quad (7) \qquad p_\mathrm{N}(x', x'' \mid x): \text{non-terminal transition} \quad (8)$$
$$p_\mathrm{T}(y \mid x): \text{terminal transition} \quad (9) \qquad p_\mathrm{S}(z \mid x): \text{termination probability} . \quad (10)$$

In analogy to PCFGs, we call this the Chomsky normal form (CNF). Any RBN may be rewritten in CNF (see Appendix A.1 for details).

**RBN Chart:** During inference, we will make use of an RBN *chart*, similar to the parse chart for PCFGs [6]. Each non-terminal variable is associated to a layer in the chart. For discrete variables, they store the actual distributions, while for continuous variables they either hold the point estimate (for MAP inference) or the parameters of the approximate distributions (for inference in Gaussian RBN). Different instances of the same template variable are identified by a subscript indicating the span of data generated from them, which also corresponds to their position in the chart (see Figure 3). Sets of variables that are generated from a specific latent non-terminal variable $x_{i:k}$ are denoted by a bold capital letter with a corresponding subscript ($\mathbf{X}_{i:k}, \mathbf{Y}_{i:k}, \mathbf{Z}_{i:k}$); omitting the subscript refers to *all* variables ($\mathbf{X}, \mathbf{Y}, \mathbf{Z}$); for $\mathbf{X}$ and $\mathbf{Z}$ this also includes the root variables $x_{0:n}$ and $z_{0:n}$, respectively. The subscripts are to be interpreted as time intervals, that is, $\mathbf{Y}_{i:i}$ is empty, $\mathbf{Y}_{0:1} = y_1$ is the first observation, $\mathbf{Y}_{n-2:n} = (y_{n-1}, y_n)$ are the last two observations etc.

**Comparison to PCFGs:** Any PCFG can be rewritten as an RBN in two different ways, which we call *abstraction* and *expansion* (see Appendix A.2 for details). Abstraction of a PCFG produces a discrete RBN with one latent non-terminal and one observed terminal variable. The resulting RBN is exactly equivalent to the original PCFG but describes the same relations in a more abstract and compact way. In contrast, *expansion* of a PCFG considers the symbols of the grammar as random variables in their own right, thereby endowing them with additional (possibly continuous) degrees

of freedom. The resulting RBN is therefore more powerful than the original PCFG. A PCFG is abstracted to a discrete RBN by defining the start/prior, transition, and structural distributions (7–10) as

$$p_\text{P}(x{=}A) = \frac{W_{S \to A}}{\sum_{A'} W_{S \to A'}} \qquad (11) \quad p_\text{N}(x'{=}B, x''{=}C \mid x{=}A) = \frac{W_{A \to BC}}{\sum_{B',C'} W_{A \to B'C'}} \quad (12)$$

$$p_\text{T}(y{=}b \mid x{=}A) = \frac{W_{A \to b}}{\sum_{b'} W_{A \to b'}} \quad (13) \quad p_\text{S}(z \mid x{=}A) = \begin{cases} \frac{\sum_{B,C} W_{A \to BC}}{\sum_X W_{A \to X}} & \text{if } z{=}\text{N} \\ \frac{\sum_b W_{A \to b}}{\sum_X W_{A \to X}} & \text{if } z{=}\text{T} \,, \end{cases} \quad (14)$$

where $S$ is the grammar's start symbol, $A, B, C$ are non-terminal symbols, $b$ is a terminal symbol, $X$ is any right-hand side of a rule, $z{=}\text{N}$ and $z{=}\text{T}$ indicate a non-terminal and terminal transition, respectively, $W_{\cdot \to \cdot}$ is the weight of the corresponding rule, and rules that do not exist in the original PCFG are taken to have zero weight. In *expansion*, the PCFG is only used to define a "skeleton" for the RBN, while the specific random variables and the concrete transition distributions need to be additionally specified. This means that the resulting RBN model is more powerful than the original PCFG, as the symbols may, for instance, be expanded to continuous random variables.

## 2.2 Inference

The two main goals of inference in RBNs are to 1) train model parameters by maximising the marginal data likelihood and to 2) compute posterior distributions or maximum posterior (MAP) estimates of the network structure and non-terminal variables. In PCFGs, both is achieved by computing inside and outside probabilities [14], which will be the starting point for our generalisation to continuous variables.

**Inside and Outside Probabilities:** We define inside and outside probabilities, $\beta$ and $\alpha$, for RBNs in analogy to how they are defined for PCFGs, the only difference being that the variables may be continuous. We thus have

$$\beta(x_{i:k}) := p(\mathbf{Y}_{i:k} \mid x_{i:k}) \qquad (15) \qquad \text{and} \qquad \alpha(x_{i:k}) := p(\mathbf{Y}_{0:i}, x_{i:k}, \mathbf{Y}_{k:n}) \,, \qquad (16)$$

where $n$ is the length of the sequence and $\mathbf{Y}$ is fixed (and therefore omitted as argument on the left-hand side). That is, $\beta(x_{i:k})$ is the marginal likelihood of generating the sub-sequence $\mathbf{Y}_{i:k}$ conditional on the respective non-terminal variable $x_{i:k}$, while $\alpha(x_{i:k})$ is the marginal likelihood of generating the two sub-sequences $\mathbf{Y}_{0:i}$ and $\mathbf{Y}_{k:n}$ as well as the non-terminal variable $x_{i:k}$. In both cases, $\beta$ and $\alpha$ are functions of the corresponding non-terminal variable with the structure and the remaining variables being marginalised out. Based on the inside and outside probabilities, the marginal data likelihood and the marginal posterior distributions over non-terminal variables are

$$p(\mathbf{Y}) = \int \beta(x_{0:n}) \, p_\text{P}(x_{0:n}) \, dx_{0:n} \qquad (17) \qquad \text{and} \qquad \widetilde{p}(x_{i:k} \mid \mathbf{Y}) = \frac{\alpha(x_{i:k}) \, \beta(x_{i:k})}{p(\mathbf{Y})} \,, \qquad (18)$$

respectively. $\widetilde{p}(x_{i:k} \mid \mathbf{Y})$ is an *unnormalised* probability distribution that specifies the probability of $x_{i:k}$ to exist via the normalisation constant $\int \widetilde{p}(x_{i:k} \mid \mathbf{Y}) \, dx_{i:k}$, while the normalised version corresponds to the marginal posterior distribution of $x_{i:k}$ for the case that it *does* exist.

Inside probabilities are recursively computed bottom-up. For an RBN in CNF we start with the base case (19) for single observations and then iterate (20) to the top of the RBN chart

$$\beta(x_{i:i+1}) = p_\text{S}(z_{i:i+1}{=}\text{T} \mid x_{i:i+1}) \, p_\text{T}(y_{i+1} \mid x_{i:i+1}) \qquad (19)$$

$$\beta(x_{i:k}) = p_\text{S}(z_{i:k}{=}\text{N} \mid x_{i:k}) \sum_{j=i+1}^{k-1} \iint p_\text{N}(x_{i:j}, x_{j:k} \mid x_{i:k}) \, \beta(x_{i:j}) \, \beta(x_{j:k}) \, dx_{i:j} \, dx_{j:k} \,. \qquad (20)$$

Outside probabilities are recursively computed top-down, while making use of the inside probabilities

$$\alpha(x_{0:n}) = p_\text{P}(x_{0:n}) \qquad (21)$$

$$\alpha(x_{j:k}) = \left[ \sum_{i=0}^{j-1} \iint p_\text{S}(z_{i:k}{=}\text{N} \mid x_{i:k}) \, p_\text{N}(x_{i:j}, x_{j:k} \mid x_{i:k}) \, \alpha(x_{i:k}) \, \beta(x_{i:j}) \, dx_{i:j} \, dx_{i:k} \right] +$$

$$\left[ \sum_{l=k+1}^{n} \iint p_\text{S}(z_{j:l}{=}\text{N} \mid x_{j:l}) \, p_\text{N}(x_{j:k}, x_{k:l} \mid x_{j:l}) \, \alpha(x_{j:l}) \, \beta(x_{k:l}) \, dx_{j:l} \, dx_{k:l} \right]. \qquad (22)$$

As for PCFGs, the two terms in (22) correspond to the possibility of $x_{j:k}$ being generated as the right or the left child, respectively. The main conceptual difference to PCFGs is that we treat the discrete structural part (marginalised out by the sums) separately from the potentially continuous variables (marginalised out by the integrals). For RBNs that are not in CNF, the equations have to be adapted accordingly (see Appendix A.3 for the general case).

**Marginalisation:** Computing the marginal data likelihood (17) and the marginal posterior distributions over non-terminal variables (18) requires to solve an exponential (w.r.t. the length $n$ of the sequence) number of nested integrals in (19–22), which is generally intractable. However, for the special case of Gaussian RBNs, we provide an adaptive closed-form approximation in Section 2.3. Moreover, marginalising *only* over the network structure for a fixed assignment of the non-terminal variables $\mathbf{X}$ is straight forward and allows for maximum posterior (MAP) inference in general RBNs.

**Maximum Posterior Inference:** For a fixed assignment of all non-terminal variables $\mathbf{X}$, we can compute the joint marginal likelihood $p(\mathbf{X}, \mathbf{Y})$ over observed terminal and latent non-terminal variables by only marginalising over the structure. This follows the same principle as above but uses the modified *joint* inside and outside probabilities

$$\widehat{\beta}_{i:k} := p(\mathbf{X}_{i:k}, \mathbf{Y}_{i:k} \,|\, x_{i:k}) \quad (23) \quad \text{and} \quad \widehat{\alpha}_{j:k} := p(\mathbf{X}_{0:j}, \mathbf{Y}_{0:j}, x_{j:k}, \mathbf{X}_{k:n}, \mathbf{Y}_{k:n}) \,, \quad (24)$$

where all variables are fixed (and therefore omitted as arguments on the left-hand side). Analogously, the joint marginal likelihood and the marginal posterior probability of $x_{i:k}$ to exist then are

$$p(\mathbf{X}, \mathbf{Y}) = \widehat{\beta}_{0:n} \, p_{\mathrm{P}}(x_{0:n}) \quad (25) \quad \text{and} \quad \widetilde{p}_{i:k} = \frac{\widehat{\alpha}_{i:k} \, \widehat{\beta}_{i:k}}{p(\mathbf{X}, \mathbf{Y})} \,, \quad (26)$$

where $\widetilde{p}_{i:k}$ is the probability of $x_{i:k}$ to exist for *this specific* assignment of $\mathbf{X}$. The corresponding equations for the recursion differ from (19–22) only in that they do not integrate out the latent non-terminal variables (see Appendix A.3.3). As before, all computations can be efficiently performed via dynamic programming. Gradients w.r.t. the variables and/or parameters are readily obtained from libraries such as `PyTorch` [26]. Optimising the values of the latent non-terminal variables $\mathbf{X}$ via gradient descent yields maximum posterior (MAP) estimates, while the structure is marginalised out. MAP estimates for the structure (i.e. the best tree) conditional on an assignment for $\mathbf{X}$ can be computed (as for PCFGs) by replacing summation with maximisation [13, 60].

There are two caveats: First, due to marginalising over multiple (exponentially many) network structures, $p(\mathbf{X}, \mathbf{Y})$ may be highly non-convex and optimising $\mathbf{X}$ via gradient descent is not guaranteed to find the global optimum. This is even the case for purely Gaussian RBNs, for which $p(\mathbf{X}, \mathbf{Y})$ is a mixture of Gaussians (one for each structure). Second, we can optimise $\mathbf{X}$ while marginalising out the structure and we can optimise the structure for a fixed assignment of $\mathbf{X}$. However, successively optimising $\mathbf{X}$ and the structure is not equivalent to *jointly* optimising both and the maximum of $p(\mathbf{X}, \mathbf{Y})$ may be unrelated to the maximum of the best structure (also see Figure 4). This means that generally, *exact joint* MAP inference over the latent variables *and* the structure is hard. For Gaussian RBNs, we provide an approximate solution below.

### 2.3 Gaussian RBNs

In a Gaussian RBN (GRBN), the prior, non-terminal, and terminal distributions are linear Gaussians and the termination probability (structural distribution) is constant

$$p_{\mathrm{P}}(x) := \mathcal{N}(x; \mu_{\mathrm{P}}, \Sigma_{\mathrm{P}}) \qquad\qquad \text{[prior]} \qquad (27)$$
$$p_{\mathrm{N}}(x', x'' \,|\, x) := \mathcal{N}(x'; x, \Sigma_{\mathrm{NL}}) \, \mathcal{N}(x''; x, \Sigma_{\mathrm{NR}}) \qquad \text{[non-terminal]} \qquad (28)$$
$$p_{\mathrm{T}}(y \,|\, x) := \mathcal{N}(y; x, \Sigma_{\mathrm{T}}) \qquad\qquad \text{[terminal]} \qquad (29)$$
$$p_{\mathrm{S}}(z{=}\mathrm{T} \,|\, x) := p_{\text{term}} \,. \qquad\qquad \text{[termination/structural]} \qquad (30)$$

For clarity, we will show all derivations for GRBNs in this basic form. For our evaluations and the application to music, we use a slightly extended version that includes linear transformations, mixtures of Gaussians, and multi-terminal transitions (Section 2.3.1). The derivations do not fundamentally change for the extended case (see Appendix A.4). In Appendix B, we show all calculations on a simple example.

**Adaptive Approximation:** If the structure of a GRBN was fixed, all variables would be jointly Gaussian distributed as in a conventional Gaussian Bayesian network [16]. However, due to the

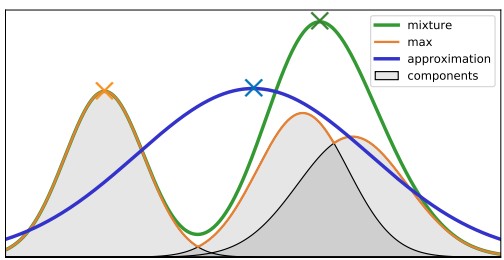

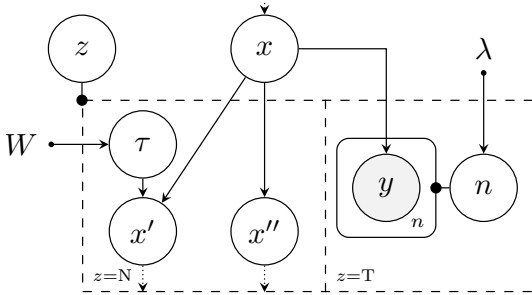

Figure 4: Three Gaussians components, the resulting mixture (green), maximum (orange), and moment-matching single Gaussian approximation (blue). Note that the maximum of the mixture ($\times$), the best component ($\times$), and the approximation ($\times$) may be unrelated.

Figure 5: Graphical model of the Gaussian RBN for modelling music. The additional transposition variable $\tau$ is marginalised out during inference; the number of jointly generated observations $n$ is uniquely determined by the location in the parse chart.

unknown structure, we effectively have a mixture of exponentially many Gaussians, one for each possible structure. While in principle all integrals can be solved analytically, the exponential growth makes exact inference intractable. Therefore, our goal is to derive a parsing strategy that retains tractability by adaptively applying local approximations to the Gaussian mixtures occurring in each recursion step. We will here focus on the simplest case of approximating the mixtures with a single Gaussian (illustrated in Figure 4, details in Appendix A.4.2), which can be efficiently computed in closed form [18, 61]. The inside and outside probabilities are thus represented by a simple Gaussian

$$\beta(x_{i:k}) \approx c_{i:k}^{(\beta)} \, \mathcal{N}(x_{i:k}; \mu_{i:k}^{(\beta)}, \Sigma_{i:k}^{(\beta)}) \qquad (31) \qquad \alpha(x_{j:k}) \approx c_{j:k}^{(\alpha)} \, \mathcal{N}(x_{j:k}; \mu_{j:k}^{(\alpha)}, \Sigma_{j:k}^{(\alpha)}) \qquad (32)$$

and this form is reestablished in each iteration by approximating the occurring mixtures. Consequently, the marginal posterior distributions over latent variables (18) are also simple Gaussians and the marginal data likelihood (17) can be computed in closed form. This approximation scheme can be extended and refined by using existing methods for approximating each Gaussian mixture by one with fewer components [62, 63].

**Marginalisation:** In (20) and (22), we have to integrate over products of Gaussian distributions to marginalise out the latent variables. To solve these integrals, we make use of the fact that the product of two Gaussians over a variable $x$ can be rewritten as [see e.g. 64]

$$\mathcal{N}(x; \mu_1, \Sigma_1) \, \mathcal{N}(x; \mu_2, \Sigma_2) = \mathcal{N}(\mu_1; \mu_2, \Sigma_1 + \Sigma_2) \, \mathcal{N}(x; \bar{\mu}, \bar{\Sigma}) \qquad (33)$$

with

$$\bar{\Sigma} := (\Sigma_1^{-1} + \Sigma_2^{-1})^{-1} \qquad \text{and} \qquad \bar{\mu} := \bar{\Sigma} \left(\Sigma_1^{-1}\mu_1 + \Sigma_2^{-1}\mu_2\right) . \qquad (34)$$

Hence, when integrating over $x$, only the first term on the rhs. of (33) remains. A detailed step-by-step derivation of all results can be found in Appendix A.4.1. With the latent variables being marginalised out, (20) and (22) become simple mixtures of Gaussians that can be easily approximated to retain the simple analytic form of the inside and outside probabilities.

**Tree Induction:** As described above, exact joint MAP inference over the continuous latent variables and the structure is generally intractable. Moreover, the maximum of the approximate posterior does not necessarily coincide with the maximum of the exact posterior or that of a particular structure (see Figure 4). Thus, first optimising $\mathbf{X}$ (based on the approximation) and then estimating the structure (conditional on the picked value of $\mathbf{X}$) may lead to arbitrarily bad results for tree induction. Therefore, we leverage the adaptive character of our approximation scheme to compute local structure estimates in each step, before loosing relevant information due to further approximations. Specifically, during the bottom-up pass for computing inside probabilities, all structures are scored by the maximum of their marginal likelihood, based on its current approximation (31). The best overall structure is then selected (as usual) in a top-down pass (see Appendix A.4.3 and our example in Appendix B).

### 2.3.1 Gaussian RBNs for Music

For the application to music, we slightly extend the basic GRBN discussed so far by introducing *transpositions* and *multi-terminal transitions* (changes in the equations highlighted in blue). The

corresponding graphical model of the RBN cell is shown in Figure 5. Furthermore, we describe how GRBNs can be applied to *categorical* data.

**Transpositions:** A transposition rotates the dimensions of the latent variable by a number of steps $\tau$ before generating the child. This is achieved by multiplying with an orthonormal transposition matrix $T_\tau$ that corresponds to the identity matrix with cyclicly rearranged columns. For the prior distribution, we assume a uniform weighting of all possible transpositions

$$p_\mathrm{P}(x) := \sum_{\tau=0}^{D-1} \frac{1}{D} \mathcal{N}(x; T_\tau \mu_\mathrm{p}, \Sigma_\mathrm{P}), \qquad \text{[prior]} \qquad (35)$$

where $D$ is the dimensionality of the data ($D = 12$ for music in 12-tone equal temperament). For the non-terminal transitions, the probability for a specific transposition is determined by the weight parameter $W$

$$p_\mathrm{N}(x', x'' \mid x) := \sum_{\tau=0}^{D-1} p(\tau \mid W) \mathcal{N}(x'; T_\tau x, \Sigma_\mathrm{NL}) \mathcal{N}(x''; x, \Sigma_\mathrm{NR}). \qquad (36)$$

Note that transpositions are only applied to the left child, because Western classical music is thought to be fundamentally goal directed [2, 8, 65]. This means that the character of a section is largely determined by how it ends (the right child), which should also be reflected in the value of the parent node. In contrast, the role of the left child is to harmonically prepare the ending (or prepare a preparation to the ending etc). We therefore allow for arbitrary transpositions in the left child and we will see below that our model indeed captures the most important type of preparation in Western classical music: the cadential dominant-tonic progression.

**Multi-Terminal Transitions:** A multi-terminal transition generates multiple observed variables from a single latent variable. The variables are generated i.i.d. and their number is governed by a Poisson distribution with rate parameter $\lambda$

$$p_\mathrm{T}(y_{i:k} \mid x_{i:k}) := \mathrm{Pois}(k - i - 1 \mid \lambda) \prod_{j=i+1}^{k} \mathcal{N}(y_j; x_{i:k}, \Sigma_\mathrm{T}). \qquad \text{[multi-terminal]} \qquad (37)$$

Multi-terminal transitions do not conform to the CNF assumed so far and we need to add the term

$$\beta(x_{i:k}) = \cdots + p_\mathrm{S}(z_{i:k}{=}\mathrm{T} \mid x_{i:k}) \, p_\mathrm{T}(y_{i:k} \mid x_{i:k}) \qquad (38)$$
$$= \cdots + p_\mathrm{term} \, p_\mathrm{T}(y_{i:k} \mid x_{i:k}) \qquad \text{[for GRBNs, see (30)]} \qquad (39)$$

to (20) in order to account for the possibility to terminate from a higher-level variable. For $k = i + 1$, this term becomes the base case (19) of an RBN in CNF.

Multi-terminal transitions account for the situation where changes in the hierarchical structure occur at a lower rate than the time series is sampled. In between the structural changes, the data is assumed to be generated from the same model, which could also be more elaborate than i.i.d. samples, as long as the relevant model parameters are captured by the RBN's latent variables.

**Categorical Data:** The observed variables of a GRBN are unconstrained real-valued, which poses a problem if the data are categorical. This situation is comparable to using Gaussian processes (GPs) [66] for classification and can be approached with similar methods. In our application to musical data, we observe one or more notes being played at any particular time and normalise these counts to obtain observations that correspond to the parameter of a categorical distribution. The natural likelihood function for this type of observations is a Dirichlet distribution. Therefore, we adapt the approach suggested in [67] for GPs, who assume a Dirichlet likelihood, which is then approximated by a Gaussian likelihood in log-space. Since an observation from a Dirichlet distribution corresponds to a normalised sample from independent Gamma distributions, each Gamma distribution can be separately approximated by a log-normal distribution, which results in a diagonal covariance matrix for the Gaussian likelihood in log-space. Matching the first and second moment yields [67]

$$\widetilde{y}_j^{(l)} = \log y_j^{(l)} - \widetilde{\Sigma}_{ll}^{(j)}/2 \qquad \text{and} \qquad \widetilde{\Sigma}_{ll}^{(j)} = \log(1/y_j^{(l)} + 1), \qquad (40)$$

where $0 < y_j^{(l)} < 1$ is the $l^\text{th}$ element (normalised count) of the $j^\text{th}$ observation, $\widetilde{y}_j^{(l)}$ is the corresponding mean of the approximate Gaussian likelihood in log-space, and $\widetilde{\Sigma}_{ll}^{(j)}$ is the $l^\text{th}$ element on the diagonal of the covariance matrix for the $j^\text{th}$ observation. We thus have to replace $\widetilde{y}_j$ and $\widetilde{\Sigma}^{(j)}$ for $y_j$ and $\Sigma_\mathrm{T}$ in (37).

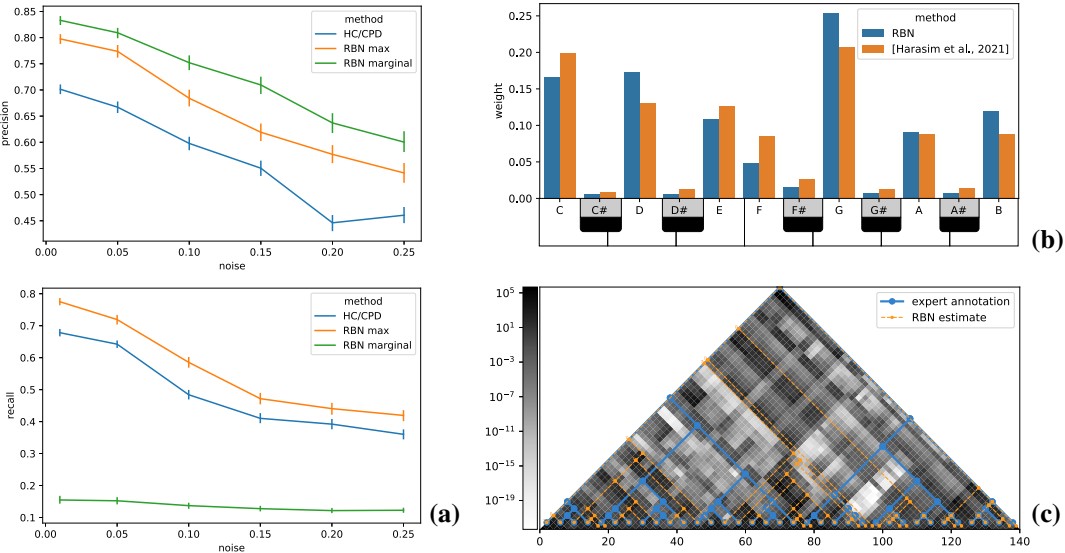

Figure 6: **(a):** Precision and recall w.r.t. the ground-truth trees of 500 sequences for different noise levels for the baseline (blue) and the maximum and marginal RBN estimates (orange, green); error bars indicate 95% confidence intervals from bootstrapping (see Appendix C.1 for technical details). **(b):** Prior mean learned by our GRBN for music in comparison to recent values from the literature [68]. **(c):** Comparison of our model (orange/grey) to an expert annotation (blue) for Johann Sebastian Bach's Prelude No. 1 in C major, BWV 846. The greyscale indicates the marginal probability of a node to exist at that particular location; the small numbers indicate the transposition in semitones for a left child; time is indicated in beats (quarter notes); the piece was divided into two-beat (half note) intervals. The plot follows the idea of *scape plots* [69, 70, 71].

## 3  Experiments

We performed a quantitative evaluation on synthetic data and applied our model to hierarchical music analysis of Bach preludes. We show that RBNs are superior to change point detection (CPD) and hierarchical clustering (HC) for tree induction and our method is able to infer fundamental harmonic principles of Western classical music. Experiments were run on a 3.6 GHz Quad-Core Intel Core i7 processor with 32GB RAM. The model parameters were trained via gradient descent on the (approximate) marginal neg-log likelihood.

### 3.1  Quantitative Evaluation on Tree Induction

We performed a quantitative evaluation on synthetic data for the task of segmenting a noisy time series and inferring the underlying tree. For comparison, we used the best-performing change point detection (CPD) method from the `ruptures` library [72] for segmenting the time series, combined with bottom-up hierarchical clustering (HC) for inferring the tree structure ("HC/CPD"). For details of the methodology, see Appendix C.1.

The evaluation results in Figure 6(a) show that the RBN tree estimates (Section 2.3) consistently outperform the one from HC/CPD, in terms of both precision and recall (and thus also in F1 measure). The marginal node probabilities show an interesting performance pattern. They excel in terms of precision, which means that a node with high marginal probability is very likely to actually exist in the tree (low false-positive rate). However, they severely underestimate the overall node probabilities, which leads to recall falling far below the baseline. This means that a node with low marginal probability may in fact occur in the tree (high false-negative rate).

We think that the poor recall measure of the marginal probabilities is primarily due to (and the downside of) a fully Bayesian treatment that quantifies uncertainty. Even if the marginal probabilities have a maximum at the correct node location, probability mass will still spread around it and be allocated to a number of less probable locations. While this is the desired behaviour of a Bayesian

method, it inevitably results in a lower recall value. The high precision value confirms that uncertainty is adequately quantified and not underestimated. That being said, the marginal probabilities provide an exceptionally rich basis for qualitative analyses. For instance, all ground-truth nodes are located at local maxima of the marginal probabilities and we can read off a number of other potential node locations, which essentially trace out the grid defined by the piece-wise constant segments (see Figure 8 in Appendix C.1).

## 3.2 Hierarchical Music Analysis

Harmonies in Western classical music exhibit a nested hierarchical structure that can be modeled by PCFGs operating on abstract chord symbols [8, 73, 10, 3]. While these grammars can be applied to expert annotations of a musical score, hierarchical music analysis from the raw note level is an unsolved problem. We trained a GRBN (Section 2.3.1) on the 24 major preludes of Johann Sebastian Bach's "Wohltemperiertes Klavier I & II" (see Appendix C.2 for technical details and complete results).

Our first major finding is that the prior mean, shown in Figure 6(b), corresponds to a major pitch profile (as could be expected from the training data) and is in excellent agreement with recent Bayesian estimates from the literature [68]. The fact that the major profile appears in the prior (i.e. as the continuous equivalent of a grammar's start symbol) shows that our model picks up fundamentally important structures from the musical data. Our second finding is that only two transpositions have non-zero weights: the identity with a weight of 78% and the fifth scale degree (7 semitones) with a weight of 22%. This corresponds to the left child being generated as the dominant of the parent and realises the most important harmonic preparation in Western classical music: the cadential dominant-tonic relation. A closer inspection of the expert analysis (Figure 10 in Appendix C.2) reveals that when considering the possible surface patterns (raw notes) of the labeled chords, most non-identity transitions can indeed be explained as (noisy) fifth transpositions. The strong weight of fifth transpositions in our model is a highly non-trivial empirical confirmation of the established music theoretical insight that Baroque music is fundamentally driven by dominant-tonic relations. While the estimated tree in Figure 6(c) fails to reproduce the large-scale structure of the expert analysis (e.g. the separation into two main parts), it accurately captures the measure-wise harmonic changes on the bottom level.

On the one hand, we see considerable room for improvement by integrating more advanced concepts, such as different modes (major/minor), diatonic in addition to chromatic transposition, or balancing of trees. On the other hand, our model was able to capture fundamental properties of Western classical music based on only 24 pieces. We therefore think that Gaussian RBNs are a highly promising approach for hierarchical music analysis from the raw note level, which should be further investigated.

## 4 Conclusion

We introduced Recursive Bayesian Networks (RBNs), a novel class of probabilistic models that unifies the strengths of probabilistic context-free grammars (PCFGs) and dynamic Bayesian networks (DBNs), generalising both model classes. We defined RBNs as a joint distribution over tree-structured Bayesian networks and their (discrete or continuous) variables and described how to perform inference over both the model structure and the variables by leveraging parsing methods for PCFGs. The provided formalisation connects with the methods for formal grammar as well as with the versatile notation for graphical models. On two data sets, we demonstrated the potential of RBNs for modelling nested hierarchical dependencies in real-valued time series and musical data. The class of RBNs represents a substantial contribution to the machine learning toolkit by unifying two of the most important approaches for modelling sequential data and bears a large potential for further development and applications.

## Acknowledgments and Disclosure of Funding

This project has received funding from the European Research Council (ERC) under the European Union's Horizon 2020 research and innovation programme under grant agreement No 760081 – PMSB. This project was conducted at the Latour Chair in Digital and Cognitive Musicology, generously funded by Mr. Claude Latour.

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
