# Appendix

*Recursive Bayesian Networks: Generalising and Unifying Probabilistic Context-Free Grammars and Dynamic Bayesian Networks*

## Table of Contents

## A   Theory

### A.1   Transformation to Chomsky Normal Form

Transforming an RBN into CNF is done analogously to the procedure for PCFGs. We assume the original RBN does not contain any epsilon productions, that is, a non-terminal variable always produces one or more other non-terminal and/or terminal variables.

**1) Eliminate terminal variables from mixed transitions:** This is done by introducing intermediate non-terminal variables. For each transition

$$p(x^{(1)}, x^{(2)}, \ldots, y^{(1)}, y^{(2)}, \ldots \mid x) , \tag{41}$$

with non-terminal variables $x^{(1)}, x^{(2)}, \ldots \in \mathcal{X}$ and terminal variables $y^{(1)}, y^{(2)}, \ldots \in \mathcal{Y}$, we introduce new non-terminal variables $x_{y^{(1)}}, x_{y^{(2)}}, \ldots$ and replace the transition by

$$p(x^{(1)}, x^{(2)}, \ldots, x_{y^{(1)}}, x_{y^{(2)}}, \ldots \mid x) , \tag{42}$$

where the new non-terminals $x_{y^{(i)}}$ replace the original terminals $y^{(i)}$. We then add new deterministic transitions

$$p(y^{(1)} \mid x_{y^{(1)}})$$
$$p(y^{(2)} \mid x_{y^{(2)}}) \tag{43}$$

$$\vdots$$

that convert each non-terminal to its equivalent terminal variable. For the newly added non-terminals, there is only a single transition and hence a degenerate structural variable that can only take a single value.

**2) Eliminate more than two latent non-terminal variables:** This is done by introducing new non-terminals that capture combinations of multiple old non-terminals. Below, we show how the number of non-terminals can be reduced by one. Applying this procedure repeatedly allows for reducing the number of non-terminals from an arbitrary number down to two, as required for CNF. A transition

$$p(x^{(1)}, x^{(2)}, \ldots, x^{(n)} \,|\, x) \tag{44}$$

that generates $n$ non-terminals $x^{(1)}, \ldots, x^{(n)}$ is rewritten as

$$p(x^{(1)}, x^{(2)}, \ldots, x^{(n-1)} \,|\, x') \, p(x', x^{(n)} \,|\, x) \,, \tag{45}$$

where we introduced the new non-terminal variable $x' = (x^{(1)}, x^{(2)}, \ldots, x^{(n-1)})$ that stores all the information from the first $n-1$ original non-terminals. The actual "work" is done by $p(x', x^{(n)} \,|\, x)$, which is the equivalent of the original $n$-fold transition. $p(x^{(1)}, x^{(2)}, \ldots, x^{(n-1)} \,|\, x')$ is a deterministic transition that just "unpacks" the information stored in $x'$. Repeating this procedure to come to only pairwise transitions corresponds to a chain of these deterministic "unpacking" operations. As above, the newly added non-terminals have only a single possible transition.

**3) Eliminate unary cycles:** Unary cycles $p_{\text{cycle}}(x' \,|\, x)$, where $x$ and $x'$ are the *same* non-terminal template variable ($x \equiv x'$), are first transformed into unary transitions to a new non-terminal variable and then eliminated as described below. We define a new non-terminal variable $\bar{x} = (x', n)$, where $n > 0$ represents the number of steps taken in the cycle before exiting it and $x'$ is the value at the moment of exiting it. The transition distribution to $\bar{x}$ is

$$p(\bar{x} \,|\, x) = p(x', n \,|\, x) = \begin{cases} p(z{\neq}\text{cycle} \,|\, x) \, p_{\text{cycle}}(x' \,|\, x) & \text{if } n = 1 \\ \int p(z{=}\text{cycle} \,|\, x'') \, p_{\text{cycle}}(x' \,|\, x'') \, p(x'', n-1 \,|\, x) \, dx'' & \text{if } n > 1 \,, \end{cases} \tag{46}$$

where in the recursive case, the variables of all intermediate steps are successively marginalised out. In practical applications, if $p(z{=}\text{cycle} \,|\, x'') < 1$, the probability of remaining in the cycle decays exponentially and the recursion can be truncated after a number of steps. If on the other hand $p(z{=}\text{cycle} \,|\, x'') \approx 1$ so that truncating is not possible, one can work with the stationary distribution of the resulting Markov chain (i.e. the Markov chain with transition distribution $p_{\text{cycle}}(x' \,|\, x)$).

The structural probability to take a transition from $x$ to $\bar{x}$ is $p(z{=}\text{cycle} \,|\, x)$, i.e. the probability of entering the cycle in the first place. The RBN cell of $\bar{x}$ is identical to that of $x$, except for the transition into the cycle, which is eliminated (the structural distribution thus has to be renormalised for the remaining transitions). The transitions use only the $x'$-component of $\bar{x}$, ignoring the $n$-component. In this way, we have expresses the state after an arbitrary number of steps in the unary cycle as a distinct value of the new non-terminal variable $\bar{x}$.

**4) Eliminate unary transitions between non-terminal variables:** Unary transitions $p_{\text{unary}}(x' \,|\, x)$, where $x$ and $x'$ are *different* non-terminal template variables, are transformed by treating $x'$ as an intermediate variable and marginalising it out. All transitions $p^{(1)}, \ldots, p^{(n)}$ from $x'$ to some other variables (terminal and/or non-terminal)

$$p^{(1)}(\ldots \,|\, x')$$
$$\vdots \tag{47}$$
$$p^{(n)}(\ldots \,|\, x')$$

are replaced by a set of new transitions $p_1, \ldots, p_n$ from $x$ directly to the respective variables, with the intermediate variable $x'$ marginalised out

$$p_1(\ldots \,|\, x) = \int p^{(1)}(\ldots \,|\, x') \, p_{\text{unary}}(x' \,|\, x) \, dx'$$

$$\vdots \tag{48}$$

$$p_n(\dots \mid x) = \int p^{(n)}(\dots \mid x') \, p_{\text{unary}}(x' \mid x) \, dx' \, .$$

The intermediate variable $x'$ and its RBN cell is eliminated if it was only reachable via $x$. The new transitions $p_1, \dots, p_n$ are merged into the cell of $x$, while the original transition $p_{\text{unary}}$ to $x'$ is removed. This requires redefining the structural distribution $p(z \mid x)$ such that the probability mass $p(z{=}\text{unary} \mid x)$ that was formerly assigned to $p_{\text{unary}}$ is now split among the new transitions $p_1$ to $p_n$ according to the structural distribution $p(z' \mid x')$ of $x'$. Specifically, for a new transition $p_i$, we define

$$p(z{=}i \mid x) := p(z{=}\text{unary} \mid x) \, p(z'{=}i \mid x') \, . \tag{49}$$

## A.2 Relation to PCFGs

As described in Section 2.1, a PCFG can be rewritten as an RBN by *abstraction* or *expansion*, where abstraction produces an equivalent RBN that describes the same relations in a more abstract and compact way, while expansion produces a more general RBN using the original PCFG as a skeleton. We describe the two procedures in detail below and use the following definition of a PCFG:

**Definition 2** (Probabilistic Context-Free Grammar). *A PCFG is a tuple* $(N, T, S, R, W)$ *of*

| | | |
|---|---|---|
| $N$ : *non-terminal symbols* | $T$ : *terminal symbols*    $S$ : *start symbol* | (50) |
| $R$ : *production rules* $\in N \times (N \cup T)^*$ | $W$ : *rule weights* . | (51) |

### A.2.1 Abstraction of a PCFG

**Theorem 1.** *A PCFG in CNF can be* abstracted *to an equivalent discrete RBN in CNF with one latent (non-terminal) template variable $x$ and one observed (terminal) template variable $y$ by defining the prior, transition, and structural distributions as*

$$p_{\text{P}}(x{=}A) = \frac{W_{S \to A}}{\sum_{A'} W_{S \to A'}} \quad (11) \qquad p_{\text{N}}(x'{=}B, x''{=}C \mid x{=}A) = \frac{W_{A \to BC}}{\sum_{B',C'} W_{A \to B'C'}} \quad (12)$$

$$p_{\text{T}}(y{=}b \mid x{=}A) = \frac{W_{A \to b}}{\sum_{b'} W_{A \to b'}} \quad (13) \qquad p_{\text{S}}(z \mid x{=}A) = \begin{cases} \dfrac{\sum_{B,C} W_{A \to BC}}{\sum_X W_{A \to X}} & \text{if } z{=}\text{N} \\[2ex] \dfrac{\sum_b W_{A \to b}}{\sum_X W_{A \to X}} & \text{if } z{=}\text{T} \, , \end{cases} \quad (14)$$

*where $A, B, C \in N$ are non-terminal symbols of the PCFG, $b \in T$ is a terminal symbol, $X \in N^2 \cup T$ is any right-hand side of a rule, $z{=}\text{N}$ and $z{=}\text{T}$ indicate a non-terminal and terminal transition, respectively, $W_{A \to X}$ is the weight of the corresponding PCFG rule, and rules that do not exist in the original PCFG are taken to have zero weight.*

To show equivalence, we need to prove that the transition probabilities from a given non-terminal symbol $A$ are the same in the original PCFG and the new RBN.

**Proof.** *In the RBN, the probability for a non-terminal transition $A \to B\,C$ is*

$$P(A \to B\,C) = p_{\text{S}}(z{=}\text{N} \mid x{=}A) \, p_{\text{N}}(x'{=}B, x''{=}C \mid x{=}A) \tag{52}$$

$$= \frac{\sum_{B',C'} W_{A \to B'C'}}{\sum_X W_{A \to X}} \frac{W_{A \to BC}}{\sum_{B',C'} W_{A \to B'C'}} \tag{53}$$

$$= \frac{W_{A \to BC}}{\sum_X W_{A \to X}} \tag{54}$$

*and that for a terminal transition $A \to b$ is*

$$P(A \to b) = p_{\text{S}}(z{=}\text{T} \mid x{=}A) \, p_{\text{T}}(y{=}b \mid x{=}A) \tag{55}$$

$$= \frac{\sum_{b'} W_{A \to b'}}{\sum_X W_{A \to X}} \frac{W_{A \to b}}{\sum_{b'} W_{A \to b'}} \tag{56}$$

$$= \frac{W_{A \to b}}{\sum_X W_{A \to X}} \, , \tag{57}$$

*which matches the corresponding probabilities in the PCFG, gained by normalising the respective weights.* $\qquad\square$

Conversely, any discrete RBN can be rewritten as a PCFG.

**Theorem 2.** *A discrete RBN with $n$ latent non-terminal template variables $x_1, \ldots, x_n$, $m$ observed terminal template variables $y_1, \ldots, y_m$, and a prior $p_\mathrm{P}(x_1)$ over $x_1$ can be rewritten as a PCFG with*

$$N := x_1 \oplus \cdots \oplus x_n \tag{58}$$

$$T := y_1 \oplus \cdots \oplus y_m \tag{59}$$

$$W_{A \to X} := \begin{cases} p_\mathrm{P}(X) & \text{if } A = S \wedge X \in x_1 \\ p(z{=}i \mid A)\, p_i(X \mid A) & \text{if a matching transition exists in the RBN} \\ 0 & \text{else,} \end{cases} \tag{60}$$

*where $\cdot \oplus \cdot$ concatenates the value ranges of the respective variables, $X \in x_i$ denote that the value $X$ is in the value range of the RBN variable $x_i$, and the second case in* (60) *requires there be a transition $p_i(x_1, \ldots, x_k \mid x_i)$ such that $A \in x_i$ and $X \in x_1 \oplus \cdots \oplus x_k$.*

#### A.2.2 Expansion of a PCFG

*Expansion* of a PCFG to an RBN uses the PCFG as a "skeleton" to define the number of template variables and the structural transitions. The domains and transitions for the variables need to be added, which results in an RBN that is more powerful than the original PCFG. Specifically, we have

$$\mathcal{X} := \{x_A \mid A \in N\} \qquad \text{and} \qquad \mathcal{Y} := \{y_b \mid b \in T\} \tag{61}$$

for the sets of latent non-terminal and observed terminal template variables and

$$p(z_A{=}X \mid x_A) = \frac{W_{A \to X}}{\sum_{X'} W_{A \to X'}} \qquad \text{with} \qquad A \in N \text{ and } X, X' \in (N \cup T)^* \tag{62}$$

for the structural transitions. Additional, we have to define the domain for each of the non-terminal and terminal variables in $\mathcal{X}$ and $\mathcal{Y}$, and for each rule $A \to X_1 X_2 \ldots$ from the original PCFG, we have to define a concrete transition distribution $p(v_{X_1}, v_{X_2}, \ldots \mid x_A)$ for the RBN (where $v_{X_1}, v_{X_2}, \ldots \in \mathcal{X} \cup \mathcal{Y}$ are non-terminal or terminal variables in the RBN, respectively, depending on whether $X_1, X_2, \ldots \in N \cup T$ are non-terminal or terminal symbols in the PCFG).

Expansion of a PCFG into an RBN seems appealing if a simple PCFG can be used to describe the *type* of variables (as opposed to their values) as well as the structure of the generative process. The actual transitions on the variables' values may then take place on a sub-symbolic/continuous level, which cannot be described by a PCFG.

### A.3 General Inside and Outside Probabilities

#### A.3.1 Inside Probabilities

The inside probability

$$\beta(x_{i:k}) = p(\mathbf{Y}_{i:k} \mid x_{i:k}) \tag{63}$$

is the probability of generating the observed terminal variables $\mathbf{Y}_{i:k}$ from the latent non-terminal variable $x_{i:k}$. This means that we need to marginalise over all possible paths of generation. Transitions may directly generate observed variables, but they may also generate lower-level non-terminals, in which case we have to recurse using the respective inside probabilities from those variables.

Let $\mathcal{T}_x \subseteq \mathcal{T}$ be the set of possible transitions from the latent non-terminal template variable $x \in \mathcal{X}$ (of which $x_{i:k}$ is one specific instantiation), with $p(z_{i:k}{=}\tau \mid x_{i:k})$ being the probability for the transition $\tau \in \mathcal{T}_x$ to be selected. This constitutes the first sum in (64) below, which marginalises over the transitions.

The transition $\tau$ generates $\eta$ new non-terminal and/or terminal variables, where $\eta$ is the arity of $\tau$. These may be located at different positions in the parse chart, depending on which part of the observed variables $\mathbf{Y}_{i:k}$ is generated from them. That is, the variables' locations in the parse chart are not known during generation and are determined in hindsight once all observed variables are generated; thus, they *are* known for parsing. We denote the respective splitting points by $j_1, \ldots, j_{\eta-1}$ (they have to fulfill the condition $i < j_1 < \ldots < j_{\eta-1} < k$) and the respective variables by

$v_{i:j_1}, \ldots, v_{j_{\eta-1}:k} \in \mathcal{X} \cup \mathcal{Y}$. The second multi-sum in (64) aggregates the probabilities of the different splitting possibilities, that is, of all valid assignments of $j_1, \ldots, j_{\eta-1}$ ($\eta - 1$ degrees of freedom). For instance, a transition of arity $\eta = 2$ has one free splitting point $j_1$ to sum over.

Some of the generated variables may be observed/terminal variables, for which nothing more needs to be done as they directly constitute the respective part of $\mathbf{Y}_{i:k}$. For the subset of non-terminal variables, which we denote by $\{v_{j:j'} \in \mathcal{X}\}$, we need to insert their respective inside probabilities and marginalise them out. This constitutes the product and multi-integral in (64).

The general form of the inside probabilities then is

$$\beta(x_{i:k}) = \sum_{\tau \in \mathcal{T}_x} p_{\mathrm{S}}(z_{i:k}{=}\tau \,|\, x_{i:k}) \sum_{i<j_1<\ldots<j_{\eta-1}<k} \cdots \sum$$
$$\int_{\{v_{j:j'} \in \mathcal{X}\}} \cdots \int p_\tau(v_{i:j_1}, \ldots, v_{j_{\eta-1}:k} \,|\, x_{i:k}) \prod_{\{v_{j:j'} \in \mathcal{X}\}} \beta(v_{j:j'}) \,. \tag{64}$$

The concrete RBNs considered in the paper have only two transitions, one non-terminal transition of arity two and one terminal transition of arity one (for CNF) or more (for the extended GRBNs used in the quantitative evaluation and for modelling music). For non-terminal transition of arity two, the multi-sum in (64) reduces to a single sum and the multi-integral to a double integral, which gives us (20). For the terminal transition, (64) simplifies to (19) or the extended version (38), respectively.

### A.3.2 Outside Probabilities

The outside probability

$$\alpha(x_{j:j'}) = p(\mathbf{Y}_{0:j}, x_{j:j'}, \mathbf{Y}_{j':n}) \tag{65}$$

is the joint probability of generating the latent non-terminal variable $x_{j:j'}$ as well as the prefix and suffix of observed terminal variables, $\mathbf{Y}_{0:j}$ and $\mathbf{Y}_{j':n}$, respectively. For this, we now have to consider all possible ways how $x_{j:j'}$ as well as the prefix and suffix could have been generated from a parent non-terminal $\bar{x}$ ($x$ and $\bar{x}$ may correspond to the same template variable or to two different ones).

Let $\mathcal{T}_x^{-1} \subseteq \mathcal{T}$ denote the set of transitions that include $x$ as one of the generated variables. Importantly, if $x$ appears multiple times in the generated variables of a particular transition, these different options of generating $x$ are represented as multiple distinct entries in $\mathcal{T}_x^{-1}$, one for each occurrence. The first sum in (66) runs over these different possibilities of generating $x$.

For a transition $\tau \in \mathcal{T}_x^{-1}$ of arity $\eta$, let $j_0, \ldots, j_\eta$ be the splitting points, including the start and end point $j_0$ and $j_\eta$ of the parent variable $\bar{x}_{j_0:j_\eta}$, which have to fulfill the condition $0 \leq j_0 < \ldots < j_\eta \leq n$ (where $n$ is the length of the sequence). One pair of adjacent splitting points $(j_m, j_{m+1})$ corresponds to the occurrence of $x_{j:j'}$, where $m$ is the position (starting at zero) at which $x$ appears in the generated variables of the particular transition $\tau$. We therefore have the additional constraints $j_m = j$ and $j_{m+1} = j'$, resulting in $\eta - 1$ remaining free indices to sum over (as for the inside probabilities above). This corresponds to the second multi-sum in (66).

The set of non-terminal variables generated from the parent $\bar{x}_{j_0:j_\eta}$, excluding $x_{j:j'}$, is denoted by $\{v_{l:l'} \in \mathcal{X}\} \setminus x_{j:j'}$. Together with the directly generated terminal variables, these generate part of the prefix and suffix, $\mathbf{Y}_{j_0:j}$ and $\mathbf{Y}_{j':j_\eta}$. The remaining prefix and suffix, $\mathbf{Y}_{0:j_0}$ and $\mathbf{Y}_{j_\eta:n}$, are generated from the parent variable $\bar{x}_{j_0:j_\eta}$. For the parent, we recurse via its outside probability $\alpha(\bar{x}_{j_0:j_\eta})$, while for the newly generated non-terminal variables (except $x_{j:j'}$), we have to use the respective inside probability $\beta(v_{l:l'})$ in (66). Additionally, we have to marginalise out the parent (first integral) and the newly generated non-terminal variables (second multi-integral).

The general outside probabilities then are

$$\alpha(x_{j:j'}) = \sum_{\tau \in \mathcal{T}_x^{-1}} \sum_{\substack{0 \leq j_0 < \ldots < j_\eta \leq n \\ j_m=j \wedge j_{m+1}=j'}} \cdots \sum \int_{\bar{x}_{j_0:j_\eta}} \int_{\{v_{l:l'} \in \mathcal{X}\} \setminus x_{j:j'}} \cdots \int p_{\mathrm{S}_a}(z_{j_0:j_\eta}{=}\tau \,|\, \bar{x}_{j_0:j_\eta}) \tag{66}$$
$$p_\tau(v_{j_0:j_1}, \ldots, x_{j:j'}, \ldots, v_{j_{\eta-1}:j_\eta} \,|\, \bar{x}_{j_0:j_\eta}) \, \alpha(\bar{x}_{j_0:j_\eta}) \prod_{\{v_{l:l'} \in \mathcal{X}\} \setminus x_{j:j'}} \beta(v_{l:l'}) \,.$$

For a non-terminal transition of arity two, as we have it in the paper, the multi-sum in (66) reduces to a single sum and $\{v_{l:l'} \in \mathcal{X}\} \setminus x_{j:j'}$ contains only a single non-terminal, the second child. Importantly, $\mathcal{T}_x^{-1}$ has two elements, one for $x_{j:j'}$ being generated as the right child and one for it being generated as the left child, which gives us (22).

### A.3.3 Joint Inside and Outside Probabilities

The joint inside and outside probabilities (23) and (24) for an RBN in CNF are computed analogously to (19–22) for the normal inside and outside probabilities, that is,

$$\widehat{\beta}_{i:i+1} = p_{\mathrm{S}}(z_{i:i+1}{=}\mathrm{T} \mid x_{i:i+1})\, p_{\mathrm{T}}(y_{i+1} \mid x_{i:i+1}) \tag{67}$$

$$\widehat{\beta}_{i:k} = p_{\mathrm{S}}(z_{i:k}{=}\mathrm{N} \mid x_{i:k}) \sum_{j=i+1}^{k-1} p_{\mathrm{N}}(x_{i:j}, x_{j:k} \mid x_{i:k})\, \widehat{\beta}_{i:j}\, \widehat{\beta}_{j:k} \tag{68}$$

$$\widehat{\alpha}_{0:n} = p_{\mathrm{P}}(x_{0:n}) \tag{69}$$

$$\widehat{\alpha}_{j:k} = \Big[\sum_{i=0}^{j-1} p_{\mathrm{S}}(z_{i:k}{=}\mathrm{N} \mid x_{i:k})\, p_{\mathrm{N}}(x_{i:j}, x_{j:k} \mid x_{i:k})\, \widehat{\alpha}_{i:k}\, \widehat{\beta}_{i:j}\Big] +$$

$$\Big[\sum_{l=k+1}^{n} p_{\mathrm{S}}(z_{j:l}{=}\mathrm{N} \mid x_{j:l})\, p_{\mathrm{N}}(x_{j:k}, x_{k:l} \mid x_{j:l})\, \widehat{\alpha}_{j:l}\, \widehat{\beta}_{k:l}\Big] . \tag{70}$$

This differs from (19–22) only by dropping the integrals and dependencies on the non-terminal variables (as their values are now fixed). Joint inside and outside probabilities for the general case are obtained from (64) and (66) analogously, i.e. again by dropping the integrals and dependencies on the non-terminal variables.

## A.4 Gaussian RBNs

In the following, we present derivations for the extended case of GRBNs, described in Section 2.3.1, which includes linear transformations $T$ for the left child. For this, we will make use of the fact that a normal distribution over a transformed variable $Tx$ can be rewritten as

$$\mathcal{N}(Tx; \mu, \Sigma) = \frac{1}{\|T\|}\mathcal{N}(x; T^{-1}\mu, T^{-1}\Sigma T^{\top^{-1}}) \tag{71}$$

$$= \mathcal{N}(x; T^{\top}\mu, T^{\top}\Sigma T) , \tag{72}$$

where $\|T\|$ is the absolute value of the determinant of $T$ and in (72) we made use of the fact that in our case, the transformation matrices are orthonormal, so that $T^{-1} = T^{\top}$ and $\|T\| = 1$.

Note that for an implementation, some of the results should be rewritten in order to minimise the number of matrix inverses that need to be taken. In particular, the identity

$$(\Sigma_1^{-1} + \Sigma_2^{-1})^{-1} = \Sigma_1(\Sigma_1 + \Sigma_2)^{-1}\Sigma_2 \tag{73}$$

is useful for the implementation, but we omit it in our derivation for clarity.

### A.4.1 Marginalisation

For the inside probability $\beta(x_{i:k})$, the integral in (20) is

$$\iint p_{\mathrm{N}}(x_{i:j}, x_{j:k} \mid x_{i:k})\, \beta(x_{i:j})\, \beta(x_{j:k})\, dx_{i:j}\, dx_{j:k}$$

$$= \sum_{\tau} w_{\tau}\, c_{i:j}^{(\beta)}\, c_{j:k}^{(\beta)} \iint \mathcal{N}(x_{i:j}; T_{\tau}x_{i:k}, \Sigma_{\mathrm{NL}})\, \mathcal{N}(x_{j:k}; x_{i:k}, \Sigma_{\mathrm{NR}})$$

$$\mathcal{N}(x_{i:j}; \mu_{i:j}^{(\beta)}, \Sigma_{i:j}^{(\beta)})\, \mathcal{N}(x_{j:k}; \mu_{j:k}^{(\beta)}, \Sigma_{j:k}^{(\beta)})\, dx_{i:j}\, dx_{j:k} \tag{74}$$

$$= \sum_{\tau} w_{\tau}\, c_{i:j}^{(\beta)}\, c_{j:k}^{(\beta)}\, \mathcal{N}(T_{\tau}x_{i:k}; \mu_{i:j}^{(\beta)}, \Sigma_{\mathrm{NL}} + \Sigma_{i:j}^{(\beta)})\, \mathcal{N}(x_{i:k}; \mu_{j:k}^{(\beta)}, \Sigma_{\mathrm{NR}} + \Sigma_{j:k}^{(\beta)}) \tag{75}$$

$$= \sum_{\tau} w_{\tau}\, c_{i:j}^{(\beta)}\, c_{j:k}^{(\beta)}\, \mathcal{N}(x_{i:k}; T_{\tau}^{\top}\mu_{i:j}^{(\beta)}, T_{\tau}^{\top}[\Sigma_{\mathrm{NL}} + \Sigma_{i:j}^{(\beta)}]T_{\tau})\, \mathcal{N}(x_{i:k}; \mu_{j:k}^{(\beta)}, \Sigma_{\mathrm{NR}} + \Sigma_{j:k}^{(\beta)}) \tag{76}$$

$$= \sum_{\tau} w_{\tau}\, c_{i:j:k}^{(\beta)}\, \mathcal{N}(x_{i:k}; \mu_{i:j:k}^{(\beta)}, \Sigma_{i:j:k}^{(\beta)}) \tag{77}$$

with

$$c_{i:j:k}^{(\beta)} := c_{i:j}^{(\beta)}\, c_{j:k}^{(\beta)}\, \mathcal{N}(T_{\tau}^{\top} \mu_{i:j}^{(\beta)}; \mu_{j:k}^{(\beta)}, T_{\tau}^{\top}[\Sigma_{\mathrm{NL}} + \Sigma_{i:j}^{(\beta)}]T_{\tau} + \Sigma_{\mathrm{NR}} + \Sigma_{j:k}^{(\beta)}) \tag{78}$$

$$\mu_{i:j:k}^{(\beta)} := \Sigma_{i:j:k}^{(\beta)} \left[ T_{\tau}^{\top} \left( \Sigma_{\mathrm{NL}} + \Sigma_{i:j}^{(\beta)} \right)^{-1} \mu_{i:j}^{(\beta)} + \left( \Sigma_{\mathrm{NR}} + \Sigma_{j:k}^{(\beta)} \right)^{-1} \mu_{j:k}^{(\beta)} \right] \tag{79}$$

$$\Sigma_{i:j:k}^{(\beta)} := \left[ \left( T_{\tau}^{\top}[\Sigma_{\mathrm{NL}} + \Sigma_{i:j}^{(\beta)}]T_{\tau} \right)^{-1} + \left( \Sigma_{\mathrm{NR}} + \Sigma_{j:k}^{(\beta)} \right)^{-1} \right]^{-1}, \tag{80}$$

where in (74) we inserted (31) and (36); in (75) we used (33) twice to rewrite the pairwise products of Gaussians over $x_{i:j}$ and $x_{j:k}$ and marginalise them out; in (76) we used (72) to rewrite the transformation; and in (77) we used (33) a third time to rewrite the resulting product as a single Gaussian over $x_{i:k}$.

For the outside probability $\alpha(x_{j:k})$, the integrals in (22) for $x_{j:k}$ being generated as the right child are

$$\iint p_{\mathrm{N}}(x_{i:j}, x_{j:k} \mid x_{i:k})\, \alpha(x_{i:k})\, \beta(x_{i:j})\, dx_{i:j}\, dx_{i:k} \tag{81}$$

$$= \sum_{\tau} w_{\tau} c_{i:k}^{(\alpha)} c_{i:j}^{(\beta)} \iint \mathcal{N}(x_{i:j}; T_{\tau}x_{i:k}, \Sigma_{\mathrm{NL}})\, \mathcal{N}(x_{j:k}; x_{i:k}, \Sigma_{\mathrm{NR}})$$

$$\mathcal{N}(x_{i:k}; \mu_{i:k}^{(\alpha)}, \Sigma_{i:k}^{(\alpha)})\, \mathcal{N}(x_{i:j}; \mu_{i:j}^{(\beta)}, \Sigma_{i:j}^{(\beta)})\, dx_{i:j}\, dx_{i:k} \tag{82}$$

$$= \sum_{\tau} w_{\tau} c_{i:k}^{(\alpha)} c_{i:j}^{(\beta)} \int \mathcal{N}(x_{j:k}; x_{i:k}, \Sigma_{\mathrm{NR}})\, \mathcal{N}(x_{i:k}; \mu_{i:k}^{(\alpha)}, \Sigma_{i:k}^{(\alpha)})\, \mathcal{N}(T_{\tau}x_{i:k}; \mu_{i:j}^{(\beta)}, \Sigma^{(1)})\, dx_{i:k} \tag{83}$$

$$= \sum_{\tau} w_{\tau} c_{i:k}^{(\alpha)} c_{i:j}^{(\beta)} \int \mathcal{N}(x_{j:k}; x_{i:k}, \Sigma_{\mathrm{NR}})\, \mathcal{N}(x_{i:k}; \mu_{i:k}^{(\alpha)}, \Sigma_{i:k}^{(\alpha)})\, \mathcal{N}(x_{i:k}; T_{\tau}^{\top} \mu_{i:j}^{(\beta)}, T_{\tau}^{\top} \Sigma^{(1)} T_{\tau})\, dx_{i:k} \tag{84}$$

$$= \sum_{\tau} w_{\tau} c_{i:k}^{(\alpha)} c_{i:j}^{(\beta)} \int \mathcal{N}(x_{j:k}; x_{i:k}, \Sigma_{\mathrm{NR}})\, \mathcal{N}(x_{i:k}; \mu^{(2)}, \Sigma^{(2)})\, \mathcal{N}(\mu_{i:k}^{(\alpha)}; T_{\tau}^{\top} \mu_{i:j}^{(\beta)}, \Sigma^{(3)})\, dx_{i:k} \tag{85}$$

$$= \sum_{\tau} w_{\tau} c_{i:k}^{(\alpha)} c_{i:j}^{(\beta)} \mathcal{N}(\mu_{i:k}^{(\alpha)}; T_{\tau}^{\top} \mu_{i:j}^{(\beta)}, \Sigma^{(3)})\, \mathcal{N}(x_{j:k}; \mu^{(2)}, \Sigma^{(4)}) \tag{86}$$

$$= \sum_{\tau} w_{\tau}\, c_{i:j:k}^{(\alpha)}\, \mathcal{N}(x_{j:k}; \mu_{i:j:k}^{(\alpha)}, \Sigma_{i:j:k}^{(\alpha)}) \tag{87}$$

with

$$\Sigma^{(1)} = \Sigma_{\mathrm{NL}} + \Sigma_{i:j}^{(\beta)} \qquad\qquad \Sigma^{(2)} = \left[ \left( \Sigma_{i:k}^{(\alpha)} \right)^{-1} + \left( T_{\tau}^{\top} \Sigma^{(1)} T_{\tau} \right)^{-1} \right]^{-1} \tag{88}$$

$$\Sigma^{(3)} = \Sigma_{i:k}^{(\alpha)} + T_{\tau}^{\top} \Sigma^{(1)} T_{\tau} \qquad \mu^{(2)} = \Sigma^{(2)} \left[ \left( \Sigma_{i:k}^{(\alpha)} \right)^{-1} \mu_{i:k}^{(\alpha)} + T_{\tau}^{\top} \left( \Sigma^{(1)} \right)^{-1} \mu_{i:j}^{(\beta)} \right] \tag{89}$$

$$\Sigma^{(4)} = \Sigma_{\mathrm{NR}} + \Sigma^{(2)} \tag{90}$$

and

$$c_{i:j:k}^{(\alpha)} = c_{i:k}^{(\alpha)}\, c_{i:j}^{(\beta)}\, \mathcal{N}(\mu_{i:k}^{(\alpha)}; T_{\tau}^{\top} \mu_{i:j}^{(\beta)}, \Sigma_{i:k}^{(\alpha)} + T_{\tau}^{\top}[\Sigma_{\mathrm{NL}} + \Sigma_{i:j}^{(\beta)}]T_{\tau}) \tag{91}$$

$$\mu_{i:j:k}^{(\alpha)} = \left[ \left( \Sigma_{i:k}^{(\alpha)} \right)^{-1} + \left( T_{\tau}^{\top}[\Sigma_{\mathrm{NL}} + \Sigma_{i:j}^{(\beta)}]T_{\tau} \right)^{-1} \right]^{-1} \left[ \left( \Sigma_{i:k}^{(\alpha)} \right)^{-1} \mu_{i:k}^{(\alpha)} + T_{\tau}^{\top} \left( \Sigma_{\mathrm{NL}} + \Sigma_{i:j}^{(\beta)} \right)^{-1} \mu_{i:j}^{(\beta)} \right] \tag{92}$$

$$\Sigma_{i:j:k}^{(\alpha)} = \Sigma_{\mathrm{NR}} + \left[ \left( \Sigma_{i:k}^{(\alpha)} \right)^{-1} + \left( T_{\tau}^{\top}[\Sigma_{\mathrm{NL}} + \Sigma_{i:j}^{(\beta)}]T_{\tau} \right)^{-1} \right]^{-1}, \tag{93}$$

where in (81) we took the constant termination probability (30) out of the integral and dropped it; in (82) we inserted (31), (32) and (36); in (83) we applied (33) to marginalise out $x_{i:j}$; in (84) we used (72) to rewrite the transformation; in (85) and (86) we used (33) twice to marginalise out $x_{i:k}$; and in (87) we rewrote the final result using (91–93). Due to the asymmetric terms in the outside probabilities, the result is somewhat more complex than for the inside probabilities.

Analogously, the integrals in (22) for $x_{j:k}$ being generated as the left child are

$$\iint p_{\mathrm{N}}(x_{j:k}, x_{k:l} \mid x_{j:l})\, \alpha(x_{j:l})\, \beta(x_{k:l})\, dx_{j:l}\, dx_{k:l} \tag{94}$$

$$= \sum_\tau w_\tau c_{j:l}^{(\alpha)} c_{k:l}^{(\beta)} \iint \mathcal{N}(x_{j:k}; T_\tau x_{j:l}, \Sigma_{\mathrm{NL}})\, \mathcal{N}(x_{k:l}; x_{j:l}, \Sigma_{\mathrm{NR}}) \tag{95}$$

$$\mathcal{N}(x_{j:l}; \mu_{j:l}^{(\alpha)}, \Sigma_{j:l}^{(\alpha)})\, \mathcal{N}(x_{k:l}; \mu_{k:l}^{(\beta)}, \Sigma_{k:l}^{(\beta)})\, dx_{j:l}\, dx_{k:l}$$

$$= \sum_\tau w_\tau c_{j:l}^{(\alpha)} c_{k:l}^{(\beta)} \int \mathcal{N}(x_{j:k}; T_\tau x_{j:l}, \Sigma_{\mathrm{NL}})\, \mathcal{N}(x_{j:l}; \mu_{j:l}^{(\alpha)}, \Sigma_{j:l}^{(\alpha)})\, \mathcal{N}(x_{j:l}; \mu_{k:l}^{(\beta)}, \Sigma^{(1')})\, dx_{j:l} \tag{96}$$

$$= \sum_\tau w_\tau c_{j:l}^{(\alpha)} c_{k:l}^{(\beta)} \int \mathcal{N}(T_\tau^\top x_{j:k}; x_{j:l}, T_\tau^\top \Sigma_{\mathrm{NL}} T_\tau)\, \mathcal{N}(x_{j:l}; \mu^{(2')}, \Sigma^{(2')}) \mathcal{N}(\mu_{j:l}^{(\alpha)}; \mu_{k:l}^{(\beta)}, \Sigma^{(3')})\, dx_{j:l} \tag{97}$$

$$= \sum_\tau w_\tau c_{j:l}^{(\alpha)} c_{k:l}^{(\beta)} \mathcal{N}(\mu_{j:l}^{(\alpha)}; \mu_{k:l}^{(\beta)}, \Sigma^{(3')})\, \mathcal{N}(T_\tau^\top x_{j:k}; \mu^{(2')}, \Sigma^{(4')}) \tag{98}$$

$$= \sum_\tau w_\tau c_{j:l}^{(\alpha)} c_{k:l}^{(\beta)} \mathcal{N}(\mu_{j:l}^{(\alpha)}; \mu_{k:l}^{(\beta)}, \Sigma^{(3')})\, \mathcal{N}(x_{j:k}; T_\tau \mu^{(2')}, T_\tau \Sigma^{(4')} T_\tau^\top) \tag{99}$$

$$= \sum_\tau w_\tau\, c_{j:k:l}^{(\alpha)} \mathcal{N}(x_{j:k}; \mu_{j:k:l}^{(\alpha)}, \Sigma_{j:k:l}^{(\alpha)}) \tag{100}$$

with

$$\Sigma^{(1')} = \Sigma_{\mathrm{NR}} + \Sigma_{k:l}^{(\beta)} \qquad \Sigma^{(2')} = \left[ (\Sigma_{j:l}^{(\alpha)})^{-1} + (\Sigma^{(1')})^{-1} \right]^{-1} \tag{101}$$

$$\Sigma^{(3')} = \Sigma_{j:l}^{(\alpha)} + \Sigma^{(1')} \qquad \mu^{(2')} = \Sigma^{(2')} \left[ (\Sigma_{j:l}^{(\alpha)})^{-1} \mu_{j:l}^{(\alpha)} + (\Sigma^{(1')})^{-1} \mu_{k:l}^{(\beta)} \right] \tag{102}$$

$$\Sigma^{(4')} = T_\tau^\top \Sigma_{\mathrm{NL}} T_\tau + \Sigma^{(2')} \tag{103}$$

and

$$c_{j:k:l}^{(\alpha)} = c_{j:l}^{(\alpha)}\, c_{k:l}^{(\beta)}\, \mathcal{N}(\mu_{j:l}^{(\alpha)}; \mu_{k:l}^{(\beta)}, \Sigma_{j:l}^{(\alpha)} + \Sigma_{\mathrm{NR}} + \Sigma_{k:l}^{(\beta)}) \tag{104}$$

$$\mu_{j:k:l}^{(\alpha)} = T_\tau \left[ (\Sigma_{j:l}^{(\alpha)})^{-1} + (\Sigma_{\mathrm{NR}} + \Sigma_{k:l}^{(\beta)})^{-1} \right]^{-1} \left[ (\Sigma_{j:l}^{(\alpha)})^{-1} \mu_{j:l}^{(\alpha)} + (\Sigma_{\mathrm{NR}} + \Sigma_{k:l}^{(\beta)})^{-1} \mu_{k:l}^{(\beta)} \right] \tag{105}$$

$$\Sigma_{j:k:l}^{(\alpha)} = \Sigma_{\mathrm{NL}} + T_\tau \left[ (\Sigma_{j:l}^{(\alpha)})^{-1} + (\Sigma_{\mathrm{NR}} + \Sigma_{k:l}^{(\beta)})^{-1} \right]^{-1} T_\tau^\top . \tag{106}$$

### A.4.2 Approximation

A Gaussian mixture distribution $p(x)$ with normalised mixture weights $c_i$, means $\mu_i$, and covariance matrices $\Sigma_i$ can be approximated with a single Gaussian as

$$\widehat{p}(x) = \mathcal{N}(x; \widehat{\mu}, \widehat{\Sigma}) \quad \text{with} \quad \widehat{\mu} = \sum_i c_i\, \mu_i \quad \text{and} \quad \widehat{\Sigma} = \sum_i c_i \left[ \Sigma_i + (\mu_i - \widehat{\mu})(\mu_i - \widehat{\mu})^\top \right] . \tag{107}$$

The approximation $\widehat{p}(x)$ matches the first and second moments of $p(x)$ and minimises the Kullback-Leibler divergence (KLD) $D_{\mathrm{KL}}[p(x) \,\|\, \widehat{p}(x)]$ [61, 18]. This direction of the KLD is the one used e.g. in expectation propagation, not the one used in e.g. variational methods [18]. That means, $\widehat{p}(x)$ will adequately represent the support and uncertainty of $p(x)$ (e.g. it will be non-zero wherever $p(x)$ is non-zero). On the other hand, a value of $x$ may have a high probability in $\widehat{p}(x)$ even though in $p(x)$ it has not (also see Figure 4).

### A.4.3 Tree Induction

Exact joint optimisation of the structure and the continuous latent variables is intractable. We therefore choose the best tree for a GRBN based the maximum of the (approximated) inside probability. Inserting (30) and (77) into (20), we have

$$\beta(x_{i:k}) = (1 - p_{\mathrm{term}}) \sum_{j=i+1}^{k-1} \sum_\tau w_\tau\, c_{i:j:k}^{(\beta)}\, \mathcal{N}(x_{i:k}; \mu_{i:j:k}^{(\beta)}, \Sigma_{i:j:k}^{(\beta)}) , \tag{108}$$

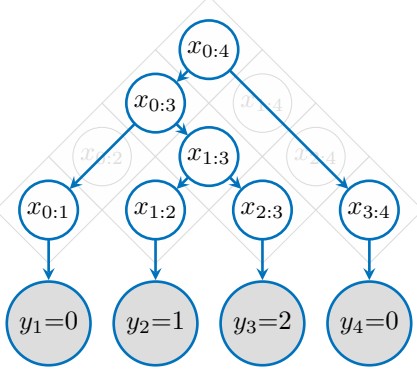

Figure 7: Parse chart for a sequence of length $n = 4$ with best tree estimate (see text for details).

which is maximised by taking the mode of the Gaussian and maximising over $j$ and (if using transpositions) $\tau$

$$\max_{(x_{i:k}, j, \tau)} w_\tau \, c_{i:j:k}^{(\beta)} \, \mathcal{N}(x_{i:k}; \mu_{i:j:k}^{(\beta)}, \Sigma_{i:j:k}^{(\beta)}) = \max_{(j, \tau)} w_\tau \, c_{i:j:k}^{(\beta)} \left| 2\pi \Sigma_{i:j:k}^{(\beta)} \right|^{-\frac{1}{2}} \tag{109}$$

If we have multi-terminal transitions (or more generally other possible transitions), we also have to maximise over the different possible transitions. For each non-terminal variable, we compute and store the best choice during bottom-up computations of the inside probabilities. Afterwards, we can construct the best tree by starting at the root node and recursively picking the best structure top-down.

## B  Example

In this section, we present the complete calculations for the inside probabilities, the tree estimate, and the marginal likelihood for a basic GRBN (no transpositions or multi-terminal transitions) on a simple example sequence of length $n = 4$ (also see Figure 7). We assume parameters

$$\mu_{\mathrm{P}} = 0 \qquad\qquad \Sigma_{\mathrm{P}} = \Sigma_{\mathrm{NL}} = \Sigma_{\mathrm{NR}} = \Sigma_{\mathrm{T}} = 1 \qquad\qquad p_{\mathrm{term}} = 1/2 \tag{110}$$

in (27–30) and a scalar sequence

$$\mathbf{Y} = (y_1, y_2, y_3, y_4) = (0, 1, 2, 0) \,. \tag{111}$$

### B.1  Preliminaries

Inside probabilities are approximated with a single Gaussian

$$\beta(x_{i:k}) \approx c_{i:k}^{(\beta)} \, \mathcal{N}(x_{i:k}; \mu_{i:k}^{(\beta)}, \Sigma_{i:k}^{(\beta)}) \,, \tag{31}$$

specified by $c_{i:k}^{(\beta)}$, $\mu_{i:k}^{(\beta)}$, and $\Sigma_{i:k}^{(\beta)}$, which are the relevant quantities to be computed.

At the bottom level, we use (19) for the base case and insert (29) and (30) to obtain

$$\beta(x_{i:i+1}) = p_{\mathrm{term}} \, \mathcal{N}(y_{i+1}; x_{i:i+1}, \Sigma_{\mathrm{T}}) \,, \tag{112}$$

where we can directly read off $c_{i:k}^{(\beta)}$, $\mu_{i:k}^{(\beta)}$, and $\Sigma_{i:k}^{(\beta)}$.

For the higher levels, we have to use (20) for the recursive case, where inserting (77) to solve the integrals in closed form gives

$$\beta(x_{i:k}) = (1 - p_{\mathrm{term}}) \sum_{j=i+1}^{k-1} c_{i:j:k}^{(\beta)} \, \mathcal{N}(x_{i:k}; \mu_{i:j:k}^{(\beta)}, \Sigma_{i:j:k}^{(\beta)}) \tag{113}$$

with parameters given by (78–80) as

$$c_{i:j:k}^{(\beta)} = c_{i:j}^{(\beta)} \, c_{j:k}^{(\beta)} \, \mathcal{N}(\mu_{i:j}^{(\beta)}; \mu_{j:k}^{(\beta)}, 1 + \Sigma_{i:j}^{(\beta)} + 1 + \Sigma_{j:k}^{(\beta)}) \tag{114}$$

$$\mu_{i:j:k}^{(\beta)} = \Sigma_{i:j:k}^{(\beta)} \left[ \left(1 + \Sigma_{i:j}^{(\beta)}\right)^{-1} \mu_{i:j}^{(\beta)} + \left(1 + \Sigma_{j:k}^{(\beta)}\right)^{-1} \mu_{j:k}^{(\beta)} \right] \tag{115}$$

$$\Sigma_{i:j:k}^{(\beta)} = \left[ \left(1 + \Sigma_{i:j}^{(\beta)}\right)^{-1} + \left(1 + \Sigma_{j:k}^{(\beta)}\right)^{-1} \right]^{-1}, \tag{116}$$

where we already inserted $\Sigma_{\mathrm{NL}} = \Sigma_{\mathrm{NR}} = 1$.

If the sum in (113) has only a single term, we immediately get

$$c_{i:k}^{(\beta)} = (1 - p_{\mathrm{term}})\, c_{i:j:k}^{(\beta)} \qquad\qquad \mu_{i:k}^{(\beta)} = \mu_{i:j:k}^{(\beta)} \qquad\qquad \Sigma_{i:k}^{(\beta)} = \Sigma_{i:j:k}^{(\beta)}. \tag{117}$$

If there is more than one term in the sum in (113), this means that there are multiple splitting options that are marginalised out and we therefore need to do two things.

First, we have to identify the best splitting option to be able to compute the tree estimate. This is done by using (109) and comparing the values of

$$\frac{c_{i:j:k}^{(\beta)}}{\sqrt{\Sigma_{i:j:k}^{(\beta)}}}, \tag{118}$$

where $\left| \Sigma_{i:j:k}^{(\beta)} \right| = \Sigma_{i:j:k}^{(\beta)}$ in the scalar case and we left out shared constant factors.

Second, we have to approximate the resulting mixture with a single Gaussian using (107), where the mixture weights have to be normalised. For the univariate/scalar case considered here, we then get

$$\mu_{i:k}^{(\beta)} = \frac{\sum_{j=i+1}^{k-1} c_{i:j:k}^{(\beta)} \mu_{i:j:k}^{(\beta)}}{\sum_{j=i+1}^{k-1} c_{i:j:k}^{(\beta)}} \tag{119}$$

$$\Sigma_{i:k}^{(\beta)} = \frac{\sum_{j=i+1}^{k-1} c_{i:j:k}^{(\beta)} \left[ \Sigma_{i:j:k}^{(\beta)} + (\mu_{i:j:k}^{(\beta)} - \mu_{i:k}^{(\beta)})^2 \right]}{\sum_{j=i+1}^{k-1} c_{i:j:k}^{(\beta)}} \tag{120}$$

$$c_{i:k}^{(\beta)} = (1 - p_{\mathrm{term}}) \sum_{j=i+1}^{k-1} c_{i:j:k}^{(\beta)}. \tag{121}$$

Finally, the marginal likelihood (17) is obtained as

$$p(\mathbf{Y}) = \int \beta(x_{0:n})\, p_{\mathrm{P}}(x_{0:n})\, dx_{0:n} \tag{122}$$

$$\approx \int c_{0:n}^{(\beta)} \mathcal{N}(x_{0:n}; \mu_{0:n}^{(\beta)}, \Sigma_{0:n}^{(\beta)}) \mathcal{N}(x_{0:n}; 0, 1)\, dx_{0:n} \tag{123}$$

$$= \int c_{0:n}^{(\beta)} \mathcal{N}(\mu_{0:n}^{(\beta)}; 0, \Sigma_{0:n}^{(\beta)} + 1) \mathcal{N}(x_{0:n}; \bar{\mu}, \bar{\Sigma})\, dx_{0:n} \tag{124}$$

$$= c_{0:n}^{(\beta)} \mathcal{N}(\mu_{0:n}^{(\beta)}; 0, \Sigma_{0:n}^{(\beta)} + 1) \tag{125}$$

where we have used (33) to rewrite the product of Gaussians and inserted $\mu_{\mathrm{P}} = 0$ and $\Sigma_{\mathrm{P}} = 1$.

The normal distribution is defined as

$$\mathcal{N}(x; \mu, \Sigma) = \frac{1}{\sqrt{2\pi|\Sigma|}} \exp\left[ -\frac{1}{2}(x - \mu)^\top \Sigma^{-1}(x - \mu) \right] \tag{126}$$

$$= \frac{1}{\sqrt{2\pi\Sigma}} \exp\left[ -\frac{1}{2} \frac{(x - \mu)^2}{\Sigma} \right], \tag{127}$$

where the second line is for the scalar case.

## B.2 Calculations

We start with the inside probabilities at the bottom level for the latent variables $x_{0:1}$, $x_{1:2}$, $x_{2:3}$, $x_{3:4}$ and from (112) we read off (without any approximations)

$$c_{i:i+1}^{(\beta)} = p_{\mathrm{term}} = 1/2 \qquad\qquad \mu_{i:i+1}^{(\beta)} = y_{i+1} \qquad\qquad \Sigma_{i:i+1}^{(\beta)} = \Sigma_{\mathrm{T}} = 1 \tag{128}$$

with

$$\mu_{0:1}^{(\beta)} = 0 \qquad \mu_{1:2}^{(\beta)} = 1 \qquad \mu_{2:3}^{(\beta)} = 2 \qquad \mu_{3:4}^{(\beta)} = 0 \,. \qquad (129)$$

Next, we compute the inside probabilities on the first level for the variables $x_{0:2}$, $x_{1:3}$, $x_{2:4}$. The only possible splitting option is for $j = i + 1$ and from (117) we get (again without approximation)

$$c_{i:i+2}^{(\beta)} = (1 - p_{\text{term}})\, p_{\text{term}}^2\, \mathcal{N}(y_{i+1}; y_{i+2}, 4), \quad \mu_{i:i+2}^{(\beta)} = (y_{i+1} + y_{i+2})/2, \quad \Sigma_{i:i+2}^{(\beta)} = 1 \quad (130)$$

and hence

$$c_{0:2}^{(\beta)} = \frac{1}{2^4 e^{\frac{1}{8}}\sqrt{2\pi}} \approx 2.20 \cdot 10^{-2} \qquad \mu_{0:2}^{(\beta)} = 0.5 \qquad \Sigma_{0:2}^{(\beta)} = 1 \qquad (131)$$

$$c_{1:3}^{(\beta)} = \frac{1}{2^4 e^{\frac{1}{8}}\sqrt{2\pi}} \approx 2.20 \cdot 10^{-2} \qquad \mu_{1:3}^{(\beta)} = 1.5 \qquad \Sigma_{1:3}^{(\beta)} = 1 \qquad (132)$$

$$c_{2:4}^{(\beta)} = \frac{1}{2^4 e^{\frac{1}{2}}\sqrt{2\pi}} \approx 1.51 \cdot 10^{-2} \qquad \mu_{2:4}^{(\beta)} = 1 \qquad \Sigma_{2:4}^{(\beta)} = 1 \,. \qquad (133)$$

Turning to the values for $x_{0:3}$ and $x_{1:4}$, we now have two terms in the sum in (113), which means that we need to evaluate the best split and approximate the mixture. The corresponding parameters of the mixture are given by (114–116) as

$$c_{i:j:k}^{(\beta)} = c_{i:j}^{(\beta)}\, c_{j:k}^{(\beta)}\, \mathcal{N}(\mu_{i:j}^{(\beta)}; \mu_{j:k}^{(\beta)}, 4)\,, \qquad \mu_{i:j:k}^{(\beta)} = (\mu_{i:j}^{(\beta)} + \mu_{j:k}^{(\beta)})/2\,, \qquad \Sigma_{i:j:k}^{(\beta)} = 1\,, \qquad (134)$$

which results in

$$c_{0:1:3}^{(\beta)} = \frac{1}{2^7 \pi e^{\frac{13}{32}}} \approx 1.66 \cdot 10^{-3} \qquad \mu_{0:1:3}^{(\beta)} = 3/4 \qquad \Sigma_{0:1:3}^{(\beta)} = 1 \qquad (135)$$

$$c_{0:2:3}^{(\beta)} = \frac{1}{2^7 \pi e^{\frac{13}{32}}} \approx 1.66 \cdot 10^{-3} \qquad \mu_{0:2:3}^{(\beta)} = 5/4 \qquad \Sigma_{0:2:3}^{(\beta)} = 1 \qquad (136)$$

and

$$c_{1:2:4}^{(\beta)} = \frac{1}{2^7 \pi e^{\frac{1}{2}}} \approx 1.51 \cdot 10^{-3} \qquad \mu_{1:2:4}^{(\beta)} = 1 \qquad \Sigma_{1:2:4}^{(\beta)} = 1 \qquad (137)$$

$$c_{1:3:4}^{(\beta)} = \frac{1}{2^7 \pi e^{\frac{13}{32}}} \approx 1.66 \cdot 10^{-3} \qquad \mu_{1:3:4}^{(\beta)} = 3/4 \qquad \Sigma_{1:3:4}^{(\beta)} = 1 \,. \qquad (138)$$

To identify the best split for each variable based on (118), we see (all variances are equal) from

$$c_{0:1:3}^{(\beta)} = c_{0:2:3}^{(\beta)} \qquad \text{and} \qquad c_{1:2:4}^{(\beta)} < c_{1:3:4}^{(\beta)}\,, \qquad (139)$$

that for $x_{0:3}$ both splits are equally well and for $x_{0:3}$ the split $x_{1:4} \to (x_{1:3}, x_{3:4})$ at $j = 3$ is better. This is intuitively clear, since generating $(y_2, y_3) = (1, 2)$ from the same non-terminal variable $x_{1:3} = 1.5$ is more likely than generating $(y_3, y_4) = (2, 0)$ from $x_{2:4} = 1$, given that in both cases the values are generated from a Gaussian with variance 1.

We approximate the mixtures with a single Gaussian with parameters given by (119–121) as

$$c_{0:3}^{(\beta)} \approx 1.66 \cdot 10^{-3} \qquad \mu_{0:3}^{(\beta)} = 1 \qquad \Sigma_{0:3}^{(\beta)} = \frac{17}{16} \qquad (140)$$

$$c_{1:4}^{(\beta)} \approx 1.58 \cdot 10^{-3} \qquad \mu_{1:4}^{(\beta)} \approx 0.869 \qquad \Sigma_{1:4}^{(\beta)} \approx 1.016 \,. \qquad (141)$$

Finally, we have the inside probability for the root variable $x_{0:4}$ with three terms in the sum in (113) with parameters

$$c_{i:j:k}^{(\beta)} = c_{i:j}^{(\beta)}\, c_{j:k}^{(\beta)}\, \mathcal{N}(\mu_{i:j}^{(\beta)}; \mu_{j:k}^{(\beta)}, 1 + \Sigma_{i:j}^{(\beta)} + 1 + \Sigma_{j:k}^{(\beta)}) \qquad (142)$$

$$\mu_{i:j:k}^{(\beta)} = \Sigma_{i:j:k}^{(\beta)} \left[ \left(1 + \Sigma_{i:j}^{(\beta)}\right)^{-1} \mu_{i:j}^{(\beta)} + \left(1 + \Sigma_{j:k}^{(\beta)}\right)^{-1} \mu_{j:k}^{(\beta)} \right] \qquad (143)$$

$$\Sigma_{i:j:k}^{(\beta)} = \left[ \left(1 + \Sigma_{i:j}^{(\beta)}\right)^{-1} + \left(1 + \Sigma_{j:k}^{(\beta)}\right)^{-1} \right]^{-1} \qquad (144)$$

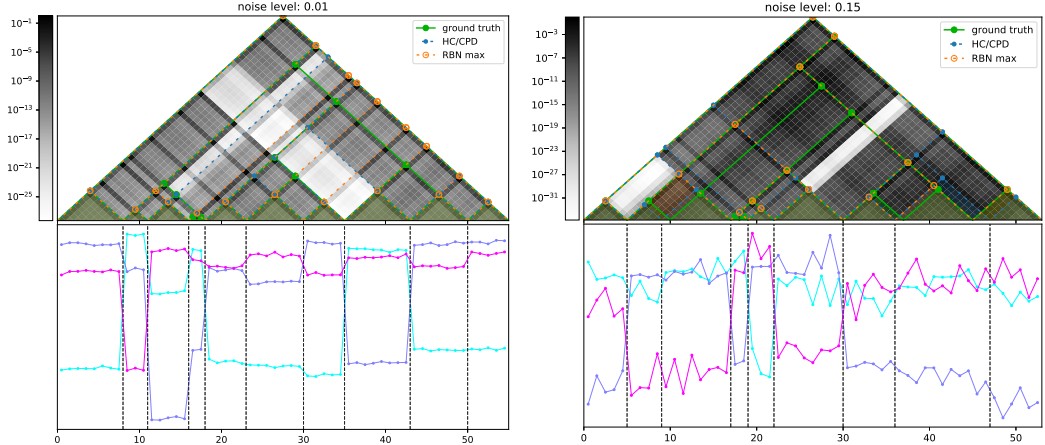

Figure 8: Example of the synthetic data used in the quantitative evaluation with noise levels of 0.01 (left) and 0.15 (right). **(bottom):** Generated three-dimensional time series; vertical dashed lines indicate the segments identified by the change point detection (CPD) method. **(top):** The ground-truth tree (green), the tree estimate from hierarchical clustering (HC) based on the CPD segmentation (blue), and the tree estimate of the RBN (orange). The grey scale indicates the marginal node probabilities based on the RBN.

and hence

$$c_{0:1:4}^{(\beta)} \approx 1.43 \cdot 10^{-4} \qquad \mu_{0:1:4}^{(\beta)} \approx 0.433 \qquad \Sigma_{0:1:4}^{(\beta)} \approx 1.004 \qquad (145)$$

$$c_{0:2:4}^{(\beta)} \approx 6.38 \cdot 10^{-5} \qquad \mu_{0:2:4}^{(\beta)} = 0.75 \qquad \Sigma_{0:2:4}^{(\beta)} = 1 \qquad (146)$$

$$c_{0:3:4}^{(\beta)} \approx 1.45 \cdot 10^{-4} \qquad \mu_{0:3:4}^{(\beta)} \approx 0.492 \qquad \Sigma_{0:3:4}^{(\beta)} \approx 1.015 \, . \qquad (147)$$

For the splitting options, (118) gives

$$\frac{c_{0:1:4}^{(\beta)}}{\sqrt{\Sigma_{0:1:4}^{(\beta)}}} \approx 1.427 \cdot 10^{-4} \qquad \frac{c_{0:2:4}^{(\beta)}}{\sqrt{\Sigma_{0:2:4}^{(\beta)}}} \approx 6.38 \cdot 10^{-5} \qquad \frac{c_{0:3:4}^{(\beta)}}{\sqrt{\Sigma_{0:3:4}^{(\beta)}}} \approx 1.439 \cdot 10^{-4} \qquad (148)$$

and we see that the split $x_{0:4} \to (x_{0:3}, x_{3:4})$ for $j = 3$ is the best one. Intuitively, this makes sense because it splits between $y_3$ and $y_4$, which is the biggest step. Not we have only minor differences between the split options, because for simplicity we have chosen our variance parameters with a value of 1, which is relatively large compared to the spread of the values. Choosing smaller variances would result in more prominent splitting preferences.

We can now construct the full tree by also picking the best split for $x_{0:3}$, which is a tie between splitting at $j = 1$ and $j = 2$, so we can choose either one (in practice one might consider random tie breaking to avoid biases due to variable order). The resulting tree is shown in Figure 7.

The parameters for the inside probability of $x_{0:4}$, given by approximating the three Gaussian mixture components, are

$$c_{0:4}^{(\beta)} \approx 1.76 \cdot 10^{-4} \qquad \mu_{0:4}^{(\beta)} = 0.515 \qquad \Sigma_{0:4}^{(\beta)} = 1.021 \, . \qquad (149)$$

Based on (125) this results in a marginal likelihood of

$$p(\mathbf{Y}) \approx 4.63 \cdot 10^{-5} \, . \qquad (150)$$

## C Experiments

### C.1 Details for Quantitative Evaluation

We performed a quantitative evaluation on synthetic data for the task of segmenting a noisy time series and inferring the underlying tree. We used the Gaussian RBN for music (Section 2.3.1) with

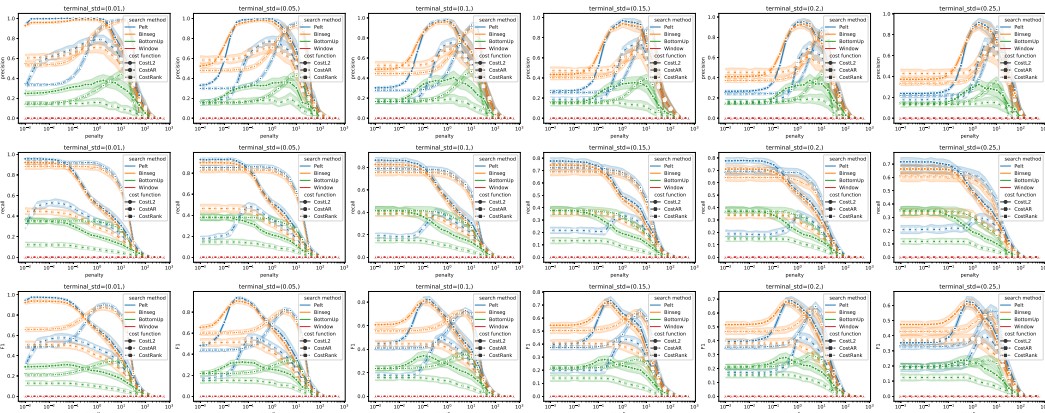

Figure 9: Grid search for best parameters and CPD method, based on the F1 score w.r.t. the ground-truth change points. The combination of `Pelt` search method with `CostL2` cost function performed best (with optimal parameters selected) in all cases.

some simplifications: 1) data were continuous and not categorical, 2) they had only three dimensions instead of twelve, 3) the prior distribution did not have any transpositions, 4) the left child could only be transposed by zero or one step. The data were sampled from this model, using zero prior mean, $\Sigma_{\mathrm{p}} = \mathbb{1}$, $\Sigma_{\mathrm{NL}} = \Sigma_{\mathrm{NR}} = \mathbb{1} \cdot 0.1^2$, $\lambda = 5$, equal weights $W_0 = W_1 = 0.5$ for transposition by zero or one, and $\Sigma_{\mathrm{T}} = \mathbb{1} \cdot \texttt{noise}^2$ with different noise levels $\{0.01, 0.05, 0.1, 0.15, 0.2, 0.25\}$. We used a terminal probability of $p_{\mathrm{term}} = 0.6$ for sampling and rejected any sequences with a length outside the range of 50–55. An example of the data is shown in Figure 8.

For comparison, we used the best-performing change point detection (CPD) method from the `ruptures` library [72] for segmenting the time series, combined with bottom-up hierarchical clustering (HC) for inferring the tree structure ("HC/CPD"). For each noise level, we selected the CPD method and parameters with best F1 score based on the ground-truth segments of 100 training sequences (see Figure 9). In HC, pairs of adjacent segments with the smallest Euclidean/L2 distance between their mean values were successively combined to construct the tree. For the RBN, all parameters were trained from scratch by minimising the marginal likelihood of the observations of only 10 training sequences (i.e. no ground-truth information and less training data than for the baseline was used), separately for each noise level.

The models were evaluated on 500 test sequences by computing their precision and recall w.r.t. the ground-truth trees. Each possible node was treated as a separate binary classification task and the results reflect the number of correctly predicted nodes. Note that due to the strongly unbalanced class distribution (many more possible node locations than actual nodes in the tree) precision and recall or the combined F1 score are the appropriate performance metrics (as opposed to e.g. accuracy). For the RBN they were computed in two different ways: 1) based on the best-tree estimate ("RBN max") and 2) based on the marginal node probabilities (18) ("RBN marginal").

Precision and recall are computed from the true positive (TP), false positive (FP), and false negative (FN) rates

$$\mathrm{recall} = \frac{\mathrm{TP}}{\mathrm{TP} + \mathrm{FN}} \tag{151}$$

$$\mathrm{precision} = \frac{\mathrm{TP}}{\mathrm{TP} + \mathrm{FP}} \tag{152}$$

$$F1 = 2 \frac{\mathrm{precision} \cdot \mathrm{recall}}{\mathrm{precision} + \mathrm{recall}} \tag{153}$$

For the single-tree estimates (baseline model and best-tree estimate from RBNs) we compared the ground-truth and estimated tree node-by-node to count correctly predicted nodes (TP), nodes that are in the prediction but not the ground-truth (FP), and nodes that are in the ground-truth but not the prediction (FN). For the marginal node probabilities, we computed the corresponding rates by counting all nodes in the ground-truth tree (TP+FN), summing the marginal probabilities over all

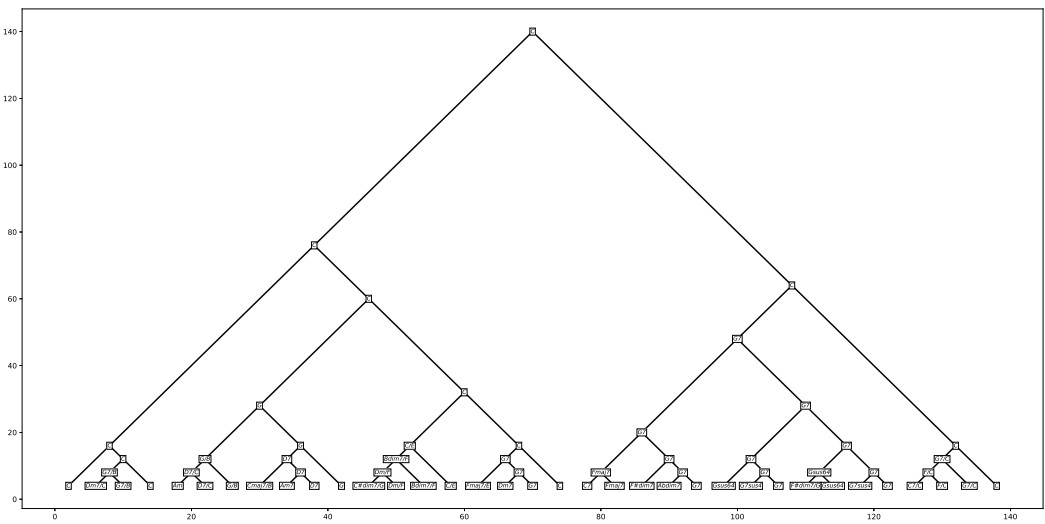

Figure 10: Harmonic analysis of Johann Sebastian Bach's Prelude No. 1 in C major, BWV 846

nodes in the ground-truth tree (TP), and summing the marginal probabilities over all possible nodes (TP+FP).

## C.2   Hierarchical Music Analysis

The scores were pre-processed by computing pitch-class distributions (PCDs), as used for the identification of musical keys [74, 75, 76], using the `pitchscapes` library [71]. We used a resolution of 70 equally spaced time slices per piece, resulting in sequences of 12-dimensional categorical distributions. The tree shown in Figure 6(c) for Johann Sebastian Bach's Prelude No. 1 in C major, BWV 846, corresponds to a harmonic expert analysis performed by the authors. Figure 10 shows the annotated tree with additional chord labels, which are provided in a simplified notation commonly used in Jazz lead sheets to be more accessible to a broad audience. In Table 1, we list the results for all 24 preludes. For a better interpretation of the model and the presented results, there are two relevant points to consider.

### C.2.1   Chromatic versus Diatonic Transposition

It is interesting to look in more detail at what musical aspects the model can or cannot represent. In a nutshell, it *can* represent chromatic transposition but *cannot* represent diatonic transposition, which has a number of consequences, as described in the following.

The transpositions of the left child perform a cyclic rotation of the corresponding probabilities in the pitch-class distribution represented by the latent variable, which corresponds to a *chromatic* transposition. This determines not only which pitch classes have a significant probability to occur (the in-scale tones) but also the specific weights. For instance, the tonic and fifth scale degree typically have the highest weights. A transposition by 5 or 7 semitones from a current major key (say C major) thus corresponds to a modulation to the sub-dominant (F major) or dominant (G major) key, respectively. This *includes* adaptation of the fourth and seventh scale degree of the target key, respectively (B→B♭ for F major; F→F♯ for G major), as well as the correct assignment of strong weights to the tonic and fifth scale degree.

However, *diatonic* transposition cannot be represented in this way. For instance, to represent a modulation from C major to A minor, the model has two options that are both far from optimal. 1) It can choose not to apply a chromatic transposition, which ensures that all in-scale tones are correctly represented (i.e. they have significant weight). This, however, means that the relative weights are not appropriate for A minor. In particular, the strong weights on the tonic and fifth scale degree are not present and, instead, the third and seventh scale degree (C and G, the former tonic and fifth scale degree) have disproportionally strong weight. Correcting these weights has to occur through the

Gaussian transitions, which can only be explained with a relatively high transition variance. 2) The second option would be to perform a chromatic transposition by 9 semitones, which ensures that the strongest weights remain on the tonic and fifth scale degree of the new key. However, three out-of-scale tones (C♯, F♯, G♯) now have a high weight, while the respective in-scale tones do not. Again, this has to be corrected for by the Gaussian transition noise at a potentially even higher cost than in the first case.

This is a highly plausible explanation for why we only see non-zero weights for the identity and (chromatic) transposition by a fifth in our experiments. Any diatonic modulations are best explained by reweighting using via Gaussian transition noise without a transposition, rather than by a chromatic transposition, which would require an even stronger reweighting (except for modulation to the sub-dominant and dominant key, which can be appropriately explained by a chromatic transposition).

### C.2.2 Chord Labels

It is important to note that the chord labels in the expert annotation convey significantly more information than just what pitch classes can be expected to occur in the respective section. For example, the very same pitch-class distribution of G–C–E could amongst others be labeled as a C major chord in second inversion, a G major chord with 64-suspension (Gsus64), or an A minor seventh chord with omitted root, which might be easily confused by a musically untrained annotator. Which of these labels is correct depends in many cases on the context, such as how a chord resolves to the next one. While these differences are important from a musical perspective (they express a different experience of the same musical events), our model was trained to only predict pitch-class distributions. Therefore, in its current state, it cannot reproduce these distinctions, but we expect future versions to significantly improve in this respect.

Table 1: Results for all major preludes in Johann Sebastian Bach's "Wohltemperiertes Klavier I & II". **(left):** Expected value of the latent variables, i.e. the mean of (18), colour-coded using a key-finding algorithm from the `pitchscapes` library [71]. **(right):** Marginal node probability, i.e. the normalisation of (18) as well as the RBN tree estimate.

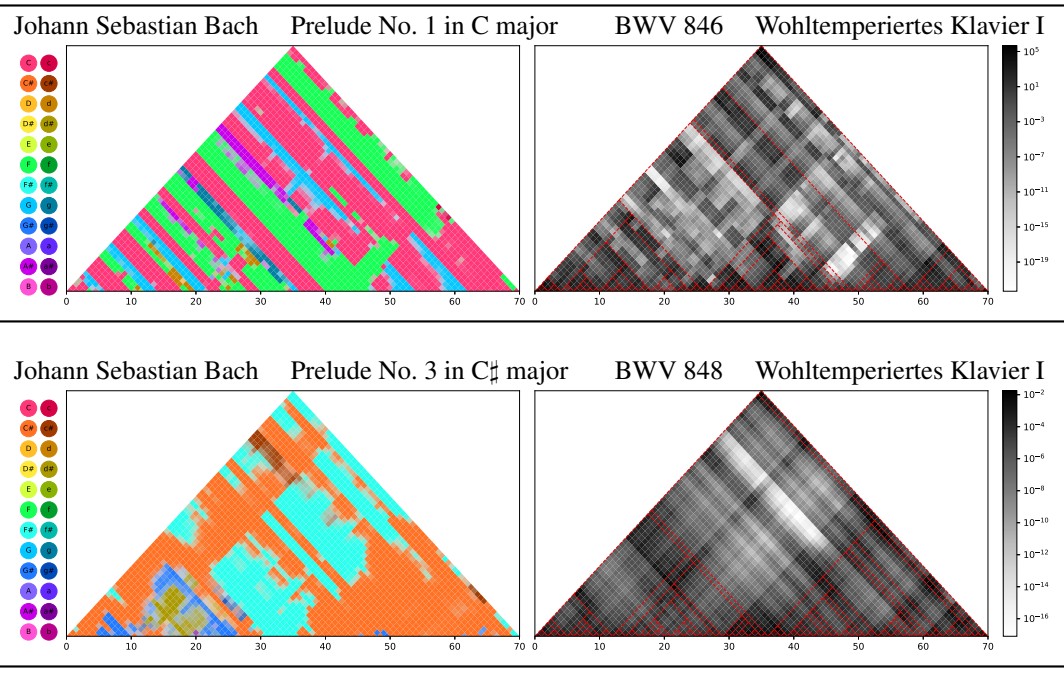

(continued on next page)

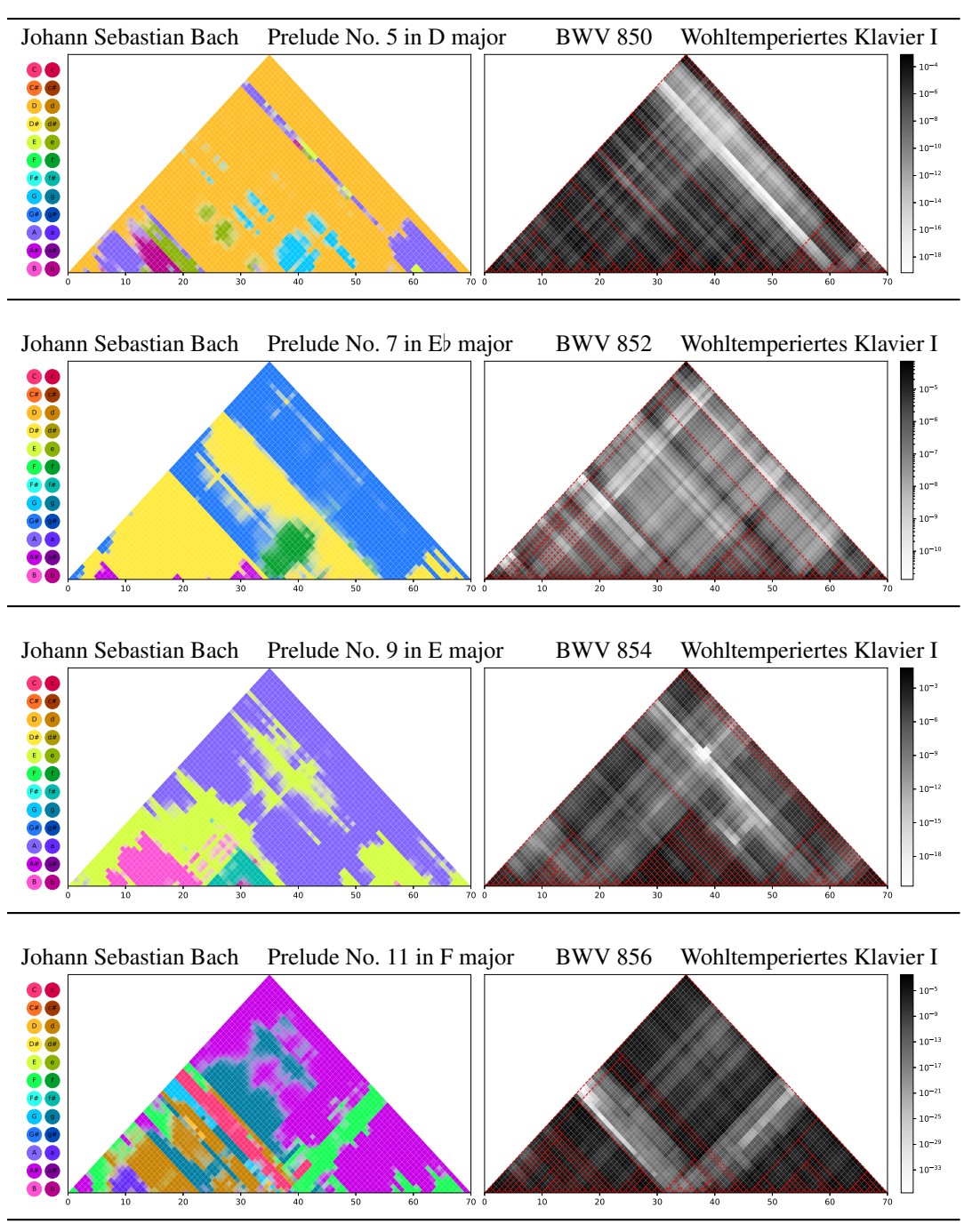

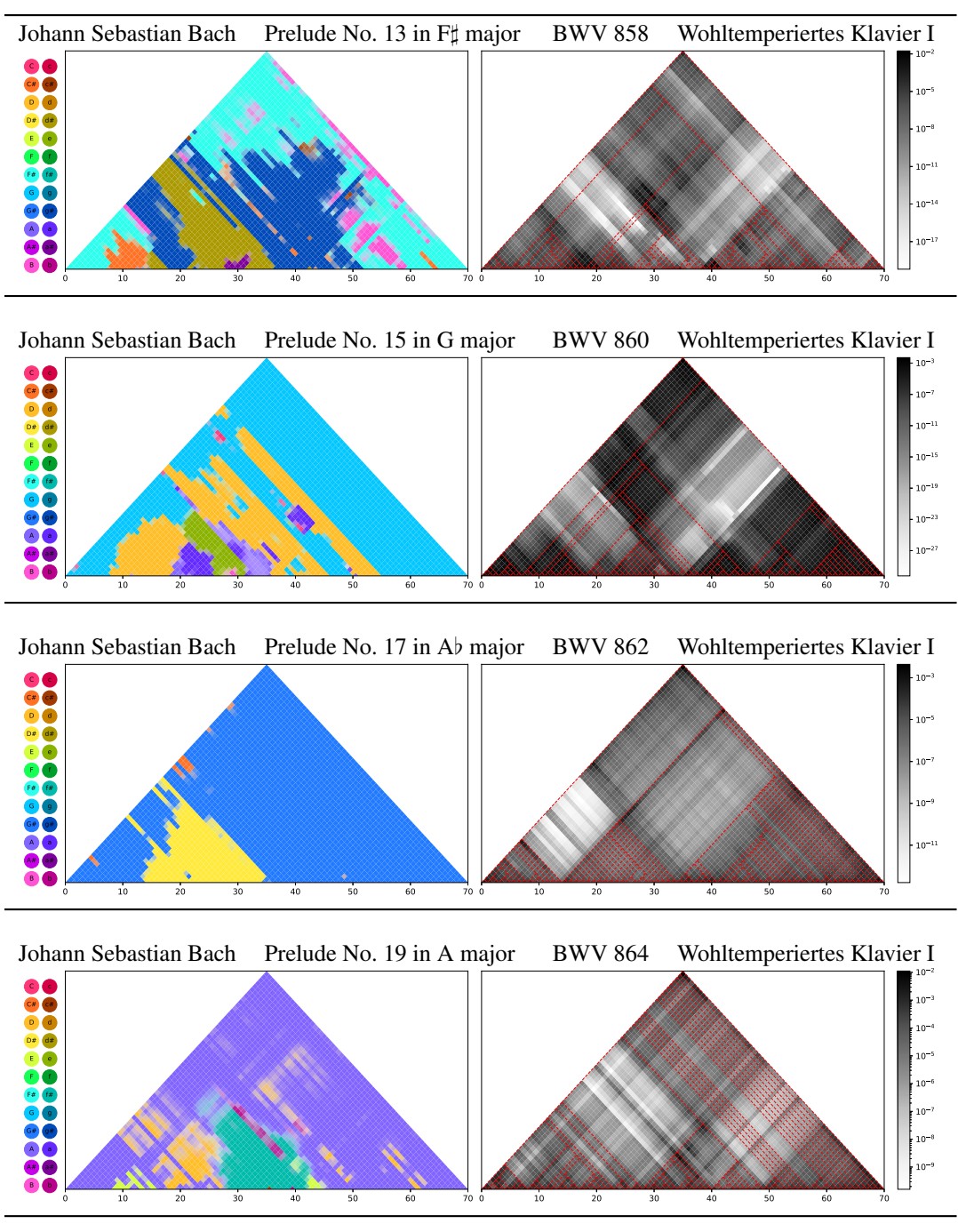

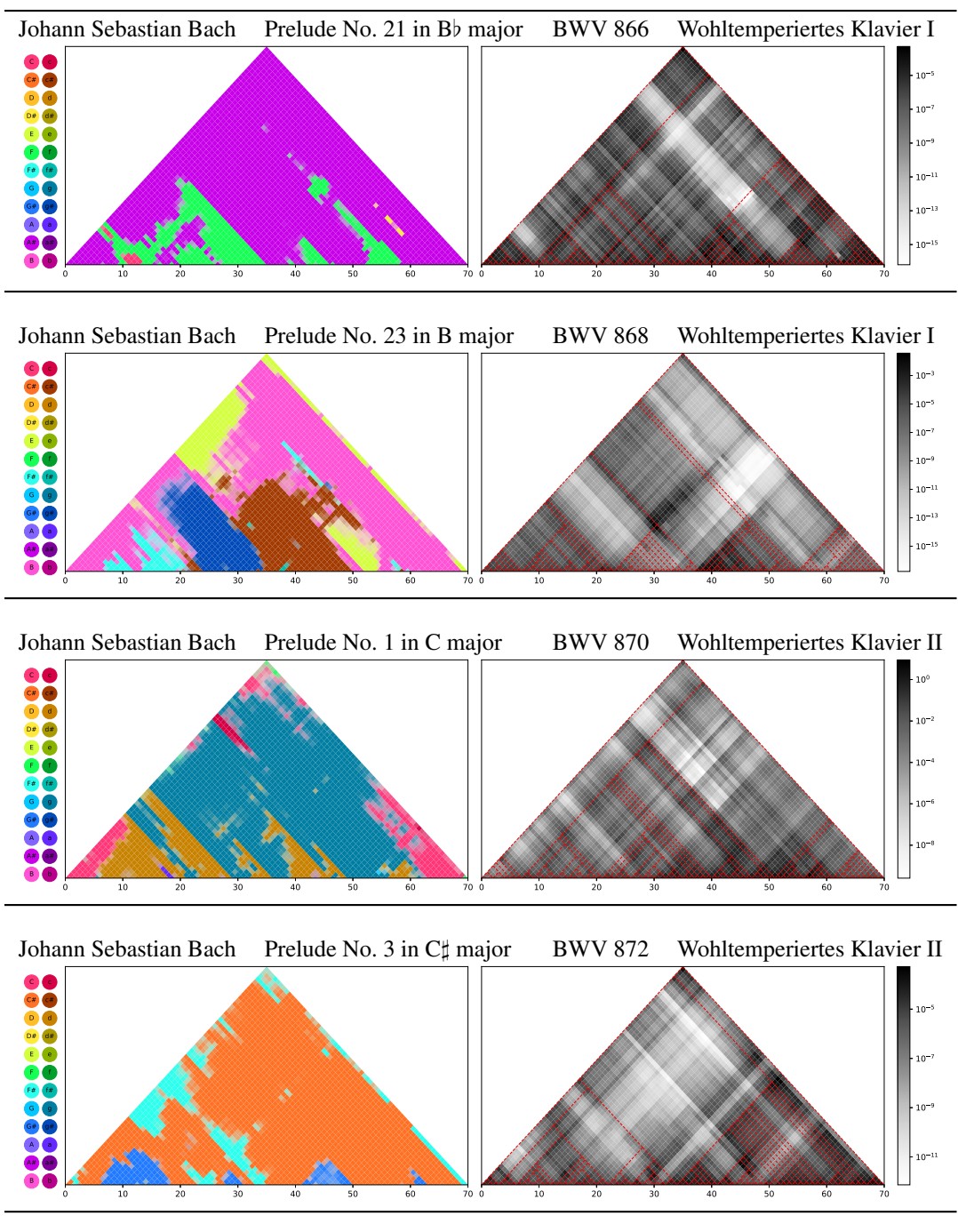

Johann Sebastian Bach    Prelude No. 21 in B♭ major    BWV 866    Wohltemperiertes Klavier I

Johann Sebastian Bach    Prelude No. 23 in B major    BWV 868    Wohltemperiertes Klavier I

Johann Sebastian Bach    Prelude No. 1 in C major    BWV 870    Wohltemperiertes Klavier II

Johann Sebastian Bach    Prelude No. 3 in C♯ major    BWV 872    Wohltemperiertes Klavier II

(continued on next page)

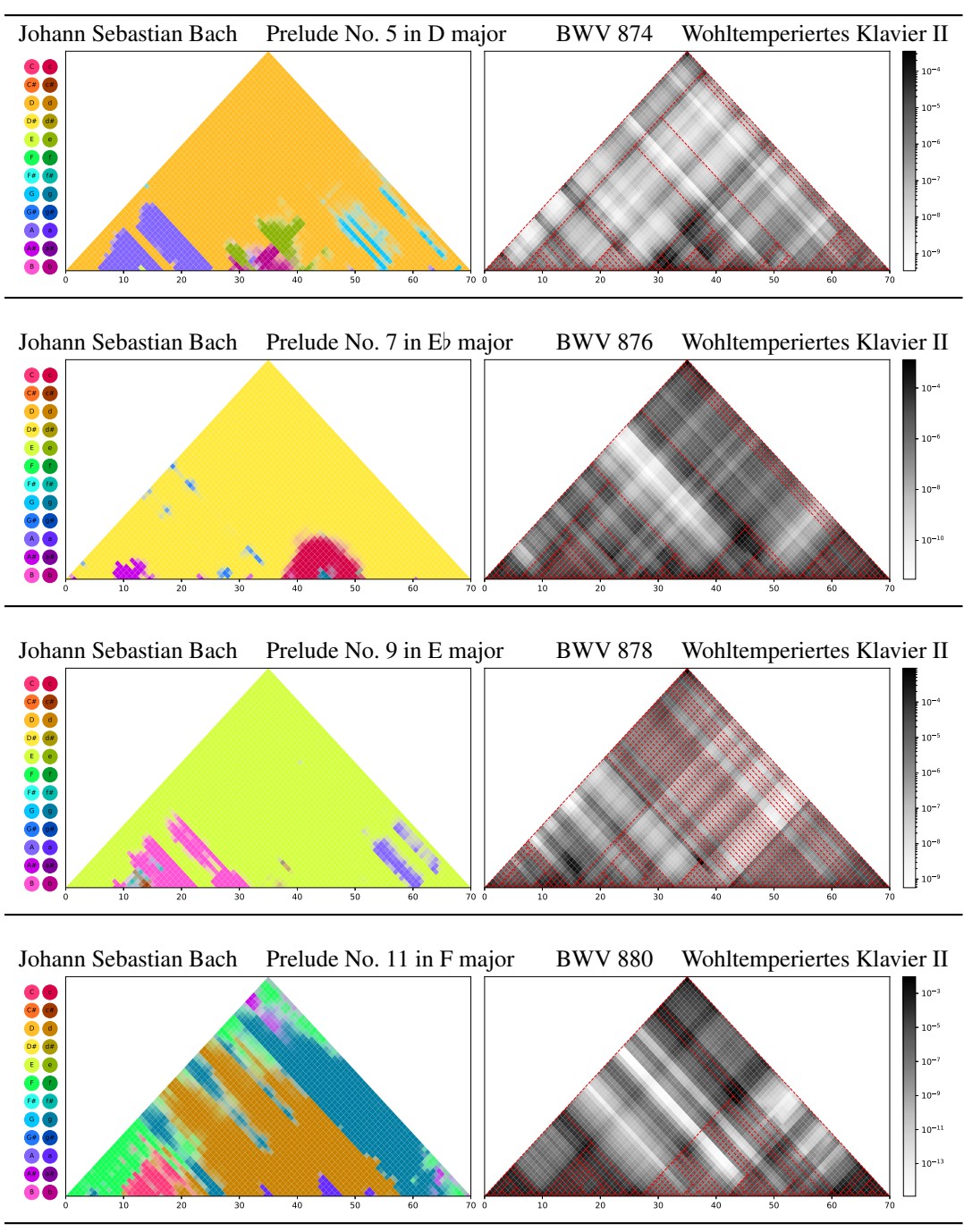

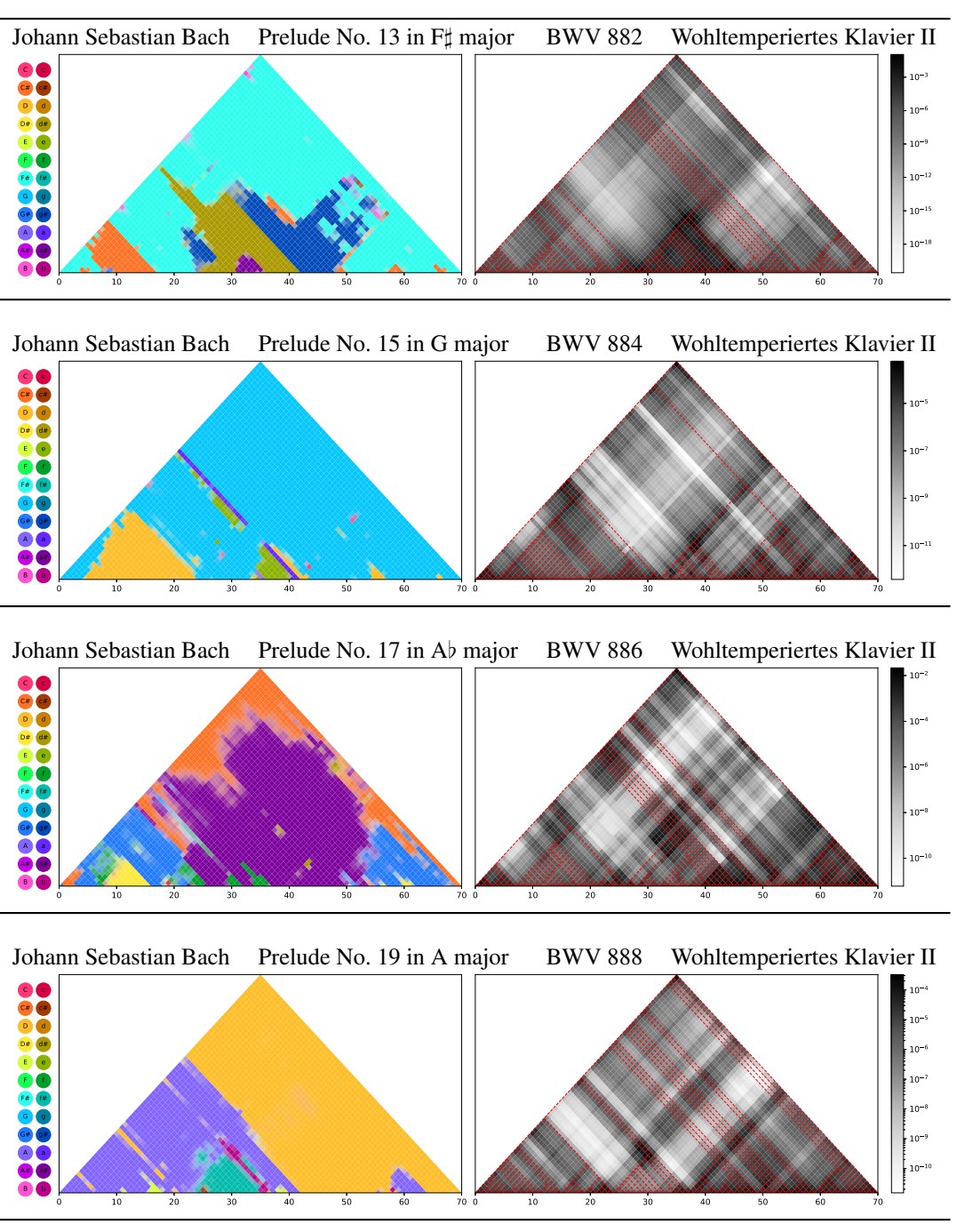

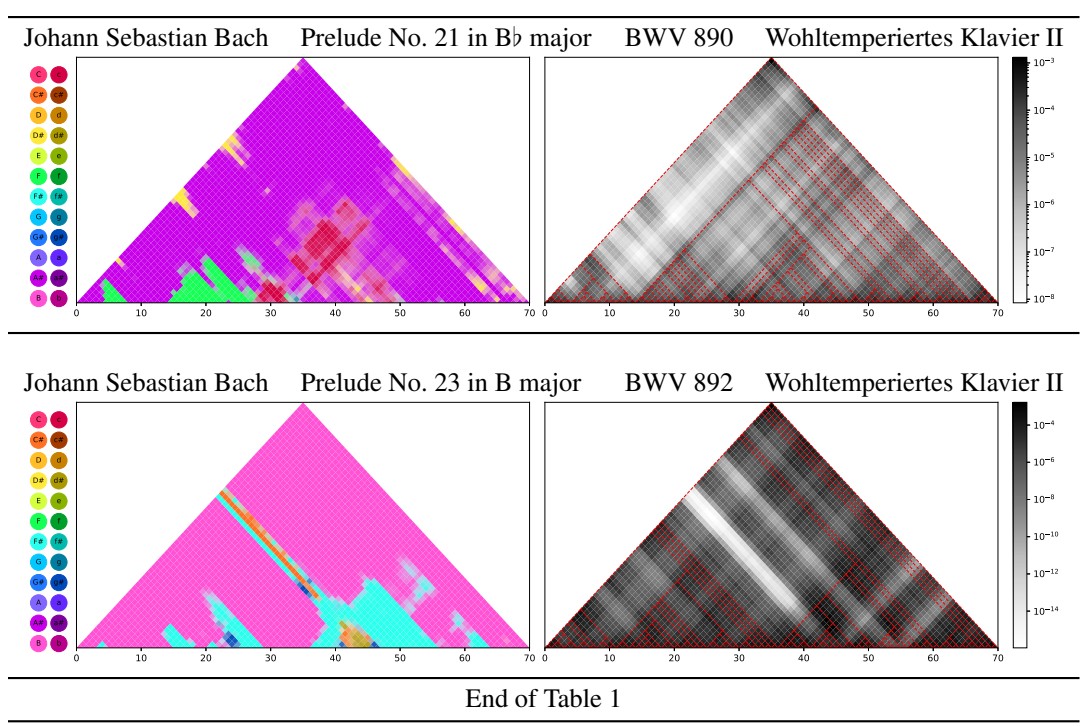