# OpenReview forum: "Recursive Bayesian Networks: Generalising and Unifying Probabilistic Context-Free Grammars and Dynamic Bayesian Networks"
_NeurIPS.cc/2021/Conference — NeurIPS 2021 Poster_

### Official Review · Reviewer_c2bx · 2021-07-13

**Rating:** 4
**Confidence:** 4

**Summary:**

The paper introduces the Recursive Bayesian Networks (RBN), which allows modelling nested hierarchical dependencies with continuous latent (non-terminal) symbols. Then, an inference procedure is derived to jointly optimize the latent structure and the model parameters. The inference procedure is the generalization of the inside-outside algorithm from PCFGs to the continuous case.

The authors also define a particular type of RBN which is based on Gaussian distributions: the Gaussian RBN (GRBN). The GBRN sets all the model continuous distributions as Gaussian distribution, obtaining a Gaussian Bayesian Network with an unknown structure. The missing structure implies a marginalization over all the possible structures that worsens the inference procedure. To keep the approach feasible, the authors introduce the “adaptive approximation” that allows approximating a mixture of Gaussian distributions with a single Gaussian.

Finally, an experimental evaluation is performed on a synthetic and a real music dataset to assess the ability of the GRBN to infer both latent structures and distributions that are consistent with the tasks.

**Limitations And Societal Impact:**

The paper has no potential negative societal impact. Nevertheless, the authors do not explicitly mention the limitations of their work.

**Main Review:**

The paper proposes an interesting enhancement of recursive probabilistic models by allowing a continuous latent state-space.

Nevertheless, I have the following concerns:

1) There is no citation/discussion on the Hidden Tree Markov Models (HTMMs) [1,2,3], which are commonly used to define a probability distribution over tree-structured data. The intrinsic recursiveness of HTMMs allows adapting on the structure of the data (that is not a priori defined and can be different for each sample). HTMMs are also related to Weighted Tree Automata (WTA), which in turn are connected to CFG. In fact, the set of all the derivation trees of a CFG is a regular tree language and therefore can be determined by a tree automata [4]. I believe that the aforementioned connections are of interest for the topic of the paper, even if they leverage a discrete latent state-space. In this regard, it is worth mentioning that Markov models for sequences with a continuous latent state-space have been already introduced in the literature [5]. Thus, I believe that the proposed RBNs are more related to the HTMMs (possibly with continuous latent state-space) rather than to the Deep Belief Networks (DBNs).  The paper would also benefit from placing the work properly in the context of the literature on latent tree estimation [6,7].

2) Even if the paper is well-written and well-organised, the paper is not always easy to understand/follow:

-	from my understanding of the RBN definition, the structural variable S associated with the non-terminal symbol A is a categorical variable whose value indicates which transition distribution associated with A should be used to expand A. Nevertheless, in the rest of the paper, the value of S is True/False (because only two transitions are used: one to generate terminal symbols and one to generate non-terminal symbols), making it difficult to understand which is the transition associated with the value of S. It would be better if the value of S is an index (or better, a label) that uniquely identify the transition distribution. This becomes even more confusing in Equation (11-14) and (24-27) since:
  + pT is used to indicate the only terminal transition distribution;
  +  pN is used to indicate the only non-terminal transition distribution;
  +  the variable S takes the values {T, F}, but the value T stands for True and not for “Terminal”. I believe that everything becomes more clear if S takes the values {T, N}, which are the label used to disambiguate the transition distributions. Also, the symbol pT denotes the set of all the transition distributions in RBN.

-	In my opinion, some aspects deserve more attention in the paper to facilitate the understanding of the proposed model, such as the training algorithm and the procedure to induce the latent tree structure. In particular, it is not clear to me how the inside-outside probabilities computed are used to train the model parameters and to infer the latent tree structures.
-	It is not clear why the transposition step is required for music data and what is its effect.

3) The experimental evaluation is too limited:
-	the paper highlights the need for probabilistic models with continuous latent state-space that also have a nested hierarchical dependency structure. Thus, I expected a comparison with models based on discrete latent variables to justify such a need. For example, why on music data do we need continuous latent variables?
-	a similar argument can be carried out for other design choices taken during the model definition. For example, a synthetic dataset could be used to assess the effectiveness of the “adaptive approximation” employed in the Gaussian RBN. Without such an analysis, it is hard to be convinced of the quality of the approximation since we lose the multi-modality of the original distribution;
-	the proposed method is compared with only one model from the literature on one non-synthetic dataset. For example, PCFG are used in Natural Language Processing to obtain the latent structure of a sentence. Thus, it would be interesting to compare the proposed approach with other models in the literature currently employed in this domain (PCGF or neural).

4)	There is no explicit discussion on the limitations of the proposed model. From my understanding, the main limitation could be the time complexity. It looks like that the inference procedure requires O(KN2) to compute the alpha-beta values, where N is the number of elements in the input sequence and K is the time required by the computation of Eq. (18-21). Also, it is worth mentioning that Eq. (18-21) require multiple marginalisations if no approximation is used.

Minor concerns and typos:
  + maybe there is a missing word after the distribution in Eq. (5);
  + in the text between Eq. (15) and Eq. (17), I believe that the symbol “α” and the symbol “a” have been confused: “α” should refer only to the probabilities computes, while “a” should refer only to the latent variables;
  + in the last paragraph of Section 3.1, there is a reference to Figure 6a which instead is related to hierarchical musical analysis. The discussion on that figure should be removed since it is not related to the figure that appears in the paper;
  + “it” instead of “is in the last row of the caption of Figure 6.

References:

[1]	M. Crouse, R. Nowak, and R. Baraniuk, “Wavelet-based statistical signal-processing using hidden markov-models,” IEEE Trans. Signal Process., vol. 46, no. 4, pp. 886–902, April 1998

[2]	M. Diligenti, P. Frasconi, and M. Gori, “Hidden tree markov models for document image classification,” IEEE Trans. Pattern Anal. Mach. Intell., vol. 25, no. 4, pp. 519–523, 2003.

[3]	D. Bacciu, A. Micheli, and A. Sperduti, “Compositional generative mapping for tree-structured data - part I: Bottom-up probabilistic modeling of trees,” IEEE Trans. Neural Netw. Learning Syst., vol. 23, no. 12, pp. 1987–2002, 2012.

[4]	H. Comon and M. Dauchet and R. Gilleron and  C. Loding and F. Jacquemard and D. Lugiez and S. Tison and M. Tommasi, “Tree Automata Techniques and Applications”.

[5]	Ainsleigh, Phillip L. Theory of continuous-state hidden Markov models and hidden Gauss-Markov models. NAVAL UNDERSEA WARFARE CENTER DIV NEWPORT RI, 2001.

[6]	Anandkumar et al Spectral Methods forLearning Multivariate Latent Tree Structure, NIPS 2011

[7]	Choi et al, Learning Latent Tree Graphical Models, JMLR 2011

=== Post Rebuttal ===

After reading through the other reviews and the Authors' responses it confirms that the paper as certainly some merit when it comes to addressing learning latent hierarchical structures with continuous latent variables. In this sense, the model is seemingly well founded and reasonable also in its strongest (motivated) approximations.

On the other hand, there seems to be little disposition towards connecting the work with existing literature and empirically comparing the approach to related (certainly less expressive) works.  A discussion of the relationships with HTMM and latent tree learning models seems needed. I reckon that HTMM do not learn a latent tree structure but are a more closely related model than DBN and HMM cited in the paper. Similarly, relationships with latent tree learning works such as [a] and latent graph structure learning [b] with deep NNs (hence inherenty continous in their representation) should be considered.

The empirical analysis is too limited to be convincing on the potential impact of the model. If the model is addressing some relevant problem there should be certainly realistic tasks where it can be applied: if not, then this reduces the impact of the contribution. Also, while I reckon that there might not be other models in literature that work on raw polyphonic note sequences, the proposed approach could be applied to NLP tasks where other latent structure learning approaches have been benchmarked on (e.g. and where a continuous latent space might not be needed), to convince the reader that the proposed method can also perform well outside the single specific dataset presented in the work (in the end, the Authors are convincing in stating that the proposed method enlarges the breadth of application of existing models, so one might want to have empirical confirmation of this aspect).

All in all, I am fine in rasing a bit my score but I am still not convinced that the current content of the paper is sufficient to support acceptance.

[a] F. Huang et al, Guaranteed Scalable Learning of Latent Tree Models, ICML 20
[b] L. Franceschi et al, Learning Discrete Structures for Graph Neural Networks, ICML 19


**Time Spent Reviewing:**

7

---

> ### Author Response · Authors · 2021-08-11
> **Point-by-point response**
>
> - Hidden Tree Markov Models (HTMMs) are used to model tree-structured data, that is, a fixed tree structure is extracted from the data. While this tree may be different for each data point, it is not treated as an unknown latent variable that needs to be marginalised out. Also the main challenge in RBNs arises from having _continuous_ latent variables, which is not addressed in HTMMs and related models. As a consequence, we see little overlap to RBNs and the problems we address in our paper.
> - We appreciate that the True/False labelling of the binary structural variable might not very intuitive and think that using {T, N} (for terminal and non-terminal) could be a good solution to improve the clarity of the presentation.
> - The model parameters are trained by computing the marginal data likelihood (via the inside probabilities), which is then optimised via gradient descent (where the gradient w.r.t. the model parameters is automatically computed via PyTorch).
> - The best tree structure computed by maximising the marginal likelihood (see our comment to reviewer uEGL).
> - If you could point us to a PCFG that is able to process the raw note input of polyphonic music, we would be happy to compare our RBN model against it. To our knowledge, this is an unsolved problem, which is one of the main motivations for exploring continuous latent representations as these have proven highly beneficial in other domains with discrete data (e.g. vector embedding for words).
> - The time complexity of parsing an RBN is analogous to that of a PCFG, that is $O(n^3)$, where $n$ is the sequence length.

---

### Official Review · Reviewer_dyhB · 2021-07-13

**Rating:** 4
**Confidence:** 3

**Summary:**

This paper is about recursive Bayesian networks, in which a PCFG generates a set of graphical models, possibly with continuous random variables. The main application explored in this paper is music analysis, which involves some interesting modifications to the basic model.

**Limitations And Societal Impact:**

I would like to see more discussion of (1) the potential limitations introduced by approximating a mixture of Gaussians by a single Gaussian; (2) the relationship of the model used to generated the data in Figure 6b to the model used to learn from the data; and (3) the apparent failure of the model in Figure 6c to recover the ground-truth tree.


**Main Review:**

Formalism

The definition of RBN allows a general CFG, but equations (18-21) assume that the CFG is in Chomsky normal form, and (35-36) extend them to rules of the form X -> a_1 ... a_m. Appendix A.1 partly describes how to convert a general CFG to Chomsky normal form, but it looks to me like two cases are missing: epsilon rules (of the form X -> \epsilon) and unary cycles (of the form X_1 -> X_2, ..., X_2 -> X_m, X_m -> X_1). Please either amend the definition or the conversion so they match up.

RBNs appear to be a special case of factor graph grammars (Chiang and Riley, NeurIPS 2020), although the present paper does far more to explore the case of inference with continuous random variables.

Model

To make inference tractable, mixtures of Gaussians are approximated by a single Gaussian, which seems to me like a pretty big approximation. Is there any way to measure how much error is introduced by this approximation? For example, would it be possible to directly measure it on a toy example?

Line 222: Are there other musical traditions where different assumptions would be more appropriate?

Experiments

In Figure 6b, what metric is used to compare trees? In Appendix B.1 it appears that the synthetic data are generated from a simplified version of the RBN model itself, so it seems not that surprising that the RBN would perform better than the baseline HC/CPD method. Please correct me if I've misunderstood something about this experiment.

In a second experiment, the model is used to analyze music by Bach. Figure 6a shows that the model learns that all the preludes are in major keys, but aren't there much simpler methods that would tell you the same thing?

It also learns that all the transpositions are either trivial or transpose up by a fifth, which is more interesting. But I'm surprised that these two add up to 100%; shouldn't there be other transpositions as well?

In Figure 6c, what is the x-axis? What do the grayscale values mean? Is it possible to show the red and/or green tree with the transpositions included? The authors write that there are "considerable deviations" between the two trees, but to me it looks like this may be an understatement -- the two trees look wildly different.

Overall, while acknowledging that it's difficult to communicate results to readers who may not know very much music theory, I believe that more could be done to validate the proposed RBN model. Perhaps experiments on a second task would be helpful.

**Time Spent Reviewing:**

2

---

> ### Author Response · Authors · 2021-08-11
> **Point-by-point response**
>
> Thanks for pointing out the missing cases of epsilon rules and unary cycles. We will adapt our description accordingly.
>
> **Factor graph grammars (FGGs)** do not account for inference with continuous variables (in fact, the paper does not even mention the possibility of continuous variables). In contrast, the generalisation of PCFGs from discrete to continuous latent variables and the derivation of tractable inference procedures for this case are core contribution of our paper. Therefore, RBNs cannot be seen as a special case of FGGs. That being said, the class of network structures described by FGGs comprises not only trees and it would be interesting to explore whether the inference methods described in our paper can be employed to generalise FGGs to account for continuous variables.
>
> **Approximation:** We agree that the approximation with a single Gaussian may potentially introduce large errors. While the Kullback–Leibler divergence (the most natural error metric for approximation of distributions) cannot be computed in closed form, there are various techniques for estimating the approximation error (Orguner & Demırekler, 2007).
>
> - Orguner U, Demırekler M (2007) Analysis of single Gaussian approximation of Gaussian mixtures in Bayesian filtering applied to mixed multiple-model estimation. International Journal of Control 80:952–967. https://doi.org/10.1080/00207170701261952
>
> **Other music traditions:** There is a plethora of musical traditions, many of which follow different principles than the Western music tradition (also many of them are not well understood and/or researched). Indian classical music might be a particularly interesting example because it also exhibits complex hierarchical structures.
>
> **Synthetic data:** Precision and recall were computed on a node-by-node basis (also see our comments to reviewer uEGL). It is correct that data were generated from a simplified RBN, which allows us to obtain the ground truth for comparison. In fact, requiring continuous latent variables in conjunction with hierarchical dependencies almost inevitably means that the underlying model must be an RBN, since this is precisely how they are defined.
>
> **Bach preludes:** The purpose of these experiments is not so much in extracting fundamentally new information from the music, but rather in demonstrating that the model picks up on relevant structures that we know are present in the data. It is correct that such a so-called key profile can be extracted by much simpler means (e.g. by simply counting the pitch-classes in a piece). But obtaining it as the continuous prior of a model with full-blown hierarchical capacities is somewhat exciting, because it emerges as the continuous equivalent of the "start symbol" that would usually be hard-coded in a PCFG for musical harmony. We agree that seeing only the identity and fifth as transpositions with non-zero weight is even more interesting. This means that the model has captured one of the most fundamental hierarchical principles of Bach's music, which is fundamentally driven by progressions along the circle of fifths.
>
> **Figure 6c:** The x-axis is time in half notes; the greyscale values indicate the marginal probability of a node to exist at the respective location. We will add this information, as well as the transpositions for the respective trees. We agree that the upper levels of the two trees have almost nothing in common. However, on the lowest levels (which contain a significant larger number of nodes), they share most of the nodes. In particular, the measure-wise harmonic changes in this particular prelude (i.e. the regular grouping of two neighbouring points) is correctly reproduced.

---

> > ### Comment · Reviewer_dyhB · 2021-08-17
> > **Figure 6c**
> >
> > > The x-axis is time in half notes; the greyscale values indicate the marginal probability of a node to exist at the respective location. We will add this information, as well as the transpositions for the respective trees. We agree that the upper levels of the two trees have almost nothing in common. However, on the lowest levels (which contain a significant larger number of nodes), they share most of the nodes. In particular, the measure-wise harmonic changes in this particular prelude (i.e. the regular grouping of two neighbouring points) is correctly reproduced.
> >
> > This is very difficult to see in the visualization, for several reasons:
> >
> > 1) The lowest levels of the tree are very small
> > 2) The red tree occludes the green tree when they do agree
> > 3) It would be much more difficult for red-green color-blind readers
> > 4) What does it mean when two branches of the tree touch? I see that the trees agree at the lowest level, but am having trouble understanding what it means because the right child of one subtree is the same as the first child of the next. In particular, is this tree just indicating that the first half of each measure is identical to the second half of each measure in this piece (except for the last three measures)?

---

> > > ### Author Response · Authors · 2021-08-18
> > > **Figure 6c**
> > >
> > > Thanks for these points, we will revise the figure accordingly.
> > >
> > > The last "split" of the RBN tree (the semi-transparently filled triangles that touch at the corners) indicate multi-terminal transitions. That is, their top corner represents the last non-terminal node, which almost always coincides with the leaf nodes of the ground truth annotation (done measure-wise). The lower left and right corner thus mark the start and end of the respective sub-sequence (generated from that non-terminal node) rather than child nodes. The fact that these mostly span two time steps means that the first and second half of each measure are generated from the same non-terminal node (which does not necessarily mean that they are perfectly identical as also the terminal transitions are noisy). We will make this visually clearer and better explain it in the figure caption.

---

> > ### Comment · Reviewer_dyhB · 2021-08-17
> > **Transpositions**
> >
> > > We agree that seeing only the identity and fifth as transpositions with non-zero weight is even more interesting.
> >
> > I think maybe I didn't make myself clear; I thought detecting identities and fifths was interesting, but at the same time I thought the fact that all other transpositions had zero weight was concerning. Your own analysis of the C major prelude in Figure 9, for example, has lots of transpositions that are not identities or fifths (right?). Is it not a problem that your model never predicts them?

---

> > > ### Author Response · Authors · 2021-08-18
> > > **Transpositions**
> > >
> > > Thanks for the clarification. We see two points that one might find suspicious about the zero weights.
> > >
> > > First, zero weights _per se_ might seem to indicate a problem with the model: In many ML models, one observes small non-zero weights that account for "background noise" and realise small non-zero probabilities for the entire data space (i.e. they avoid diverging log-likelihoods for outliers in the data). In Gaussian RBNs, this is already accounted for by the continuous Gaussian transitions (which have infinite support), so zero weights are not problematic _per se_.
> > >
> > > Second (and this seems to be your main concern), harmonic analyses (incl. our own for the C major prelude) do not only contain identities and fifth transpositions. This means that there is some mismatch between the ground truth and what the model has learned. The question is how large this mismatch actually is and whether it indicates a problem. In short, we think the mismatch is actually quite small (most transitions in the analysis _are_ identities or fifth transpositions, but sometimes "in disguise") and the fact that our Gaussian RBN only relies on these two underlines the strengths of a mixed symbolic/sub-symbolic approach.
> > >
> > > The mismatch is indicative of a problem that comes with a purely symbolic approach to music analysis: Expert analyses make use of a vast (discrete) vocabulary with many of the labels representing subtle differences that require years of training to identify correctly. A simple example would be the notes G–C–E, which could (amongst others) be labeled as a C major chord in second inversion, a G major chord with 64-suspension (Gsus64), or an A minor seventh chord with omitted root. Yet, on the raw note level they are identical and which of these labels is correct depends on things like the harmonic context and voice-leading properties (e.g. a Gsus64 tends to resolve with stepwise voice leading to a normal G major chord). This illustrates some of the complications and is one of the reasons why a purely symbolic approach (e.g. using PCFGs) is challenging and possibly even questionable per se.
> > >
> > > In our analysis, most non-identity transitions turn out to be (noisy) fifth transpositions when accounting for these ambiguities in the labels. In left-most derivation order (ignoring identities and bass notes "/") we have for the first half of the piece: C --> G7 --> Dm7 [2xfifth]; C --> G --> D7 --> Am [3xfifth]; G --> D7 --> Cmaj7$\approx$Am [2xfifth]; C --> Bdim7$\approx$G7 --> Dm --> C#dim7$\approx$A7 [3xfifth]; C --> G7 --> Fmaj7$\approx$Dm7 [2xfifth]; G7 --> Dm7 [1xfifth]. Here X$\approx$Y denotes chords that have a very similar surface pattern (i.e. raw notes) and might easily be confused by an unexperienced human analyst (like G–C–E in our example above). This means that in the first half, virtually _all_ transitions can be explained as fuzzy fifth transpositions. The seconds half mostly follows a similar pattern (with some exceptions that are somewhat more noisy when explained via fifth transpositions).
> > >
> > > Of course, we hope that future versions of our model will also be able to reproduce these more subtle differences in harmonic annotations. But explaining the harmonies in Bach's preludes as "hierarchical fuzzy fifth transpositions" is an astonishingly appropriate summary and the best one we can imagine such a relatively simple model to produce. (That being said, of course, any music theorist inevitably cringes at such a gross oversimplification – just like any computer scientist cringes at the vagueness of many music theoretic concepts.)
> > >
> > > (We also noticed that the x-scale is different in Figure 6c and Figure 9, which will be fixed.)

---

### Official Review · Reviewer_k7kf · 2021-07-14

**Rating:** 4
**Confidence:** 1

**Summary:**

Good contribution, but very narrow domain for empirical validation. The paper presents Recursive BNs as an extension of classical DBNs based on PCFGs. The definition is almost straightforward, while inference might be critical. Yet, reasonable approximations are derived for the Gaussian case. The model is applied to music data (two data sets). Results are promising.

**Ethical Concerns:**

No ethical issues.

**Limitations And Societal Impact:**

I think so.

**Main Review:**

The class of models presented by the author(s) sound well motivated and perfectly reasonable. Yet, their practical advantages compared to simpler models are hard to evaluated with the current experiments. The considered domain (music) is very specific, while other contributions in the field also consider data about DNA, climatological, speech and many other, possibly more relavant, applications. For this simple reason, I think the paper might be not enough for being accepted at NeurIPS 2021.

**Time Spent Reviewing:**

One hour.

---

### Official Review · Reviewer_uEGL · 2021-07-21

**Rating:** 5
**Confidence:** 3

**Summary:**

This paper presents an approach for hierarchical modelling where the structure of a model is a part of the model itself (similar e.g. to PCFGs). It is called Recursive Bayesian Networks (RBNs). An RBN model has a prior over its structure, and can consist of continuous and discrete latent (non-latent) and observable (terminal) variables. The authors consider this as an extension (generalisation) over PCFGs (and DBNs).

To explore the joint of the model, for the general case the authors extend the inside-outside algorithm for maximum a posteriori estimates of latent variables.

The authors also present a special case of RBNs, Gaussian RBNs (with some extensions) for hierarchical models (where structure inference is possible) with continuous variables. For this special case, the submission presents an analytic approach (that also uses inside and outside probabilities) for approximate estimation of a marginal likelihood and a marginal posterior.

The authors present two empirical experiments: (a) with some synthetic data compared to two other approaches; (b) with musical data for hierarchical music analysis (the analysis includes a comparison to the expert annotation performed by the authors).


**Limitations And Societal Impact:**

Acknowledged (at least partially) by the authors:

A.1. It is unclear whether a continuous optimisation approach described in the section in lines 161-173 would be generally efficient.

A.2. How reliable/limited is the approximation by a simple Gaussian as described in lines 191-193?

A.3. The second empirical experiment results show that the model has not been able to match the expert annotations.

A.4. The first experiment is based on synthetic data produced by a modified version of the model.

A.5. The second experiment is using some annotations that (presumably) have not been independently reviewed (?).

Not acknowledged (to the best of my understanding) by the authors:

B.1. Empirical evaluation concerns only continuous variables (and not e.g. a combination of both continuous and discrete variables, which is the general case of RBNs).

B.2. The general approach of RBNs, with continuous and discrete nodes (as per the definition of the RBN), is not tested fully. Only the case of Gaussian RBNs, with additional features, is tested.

B.3. Experiments are limited: no comparison to existing methods on existing (i.e. independently of the paper) datasets (synthetic or not).

*

Societal impact: there are potential huge positive impacts of the paper, and that is really appreciated. An obvious negative social impact might be if e.g. approximations do not fit a particular application (e.g. because of using a single Gaussian) and that might lead (unexpectedly) to (e.g. significantly) reduced accuracy (that might be not acceptable for some cases of a particular application); or an inference might not converge (and it might be not clear whether it has converged or not, and the model can be used nevertheless).


**Main Review:**

This is an extensive work and it covers quite a few things. Thanks a lot to the authors for it. This is important for the field of generative models and inference.

The paper is written generally well.

Some relevant related work is covered. Based on that, generally the submission overall can be considered original. Some potentially relevant related work areas have not been covered, including: learning Bayesian networks for continuous/discrete variables (e.g. potentially ‘ “Ideal Parent” Structure Learning for Continuous Variable Bayesian Networks’ and ‘Structured Priors for Structure Learning’); and program synthesis/program induction of probabilistic programs that represent generative models.

Clarity:

A. There are a lot of parts in this paper, but some parts are not covered in detail for full clarity (some parts are covered in the appendix, but some parts are not covered at all, to the best of my understanding). That limits an (easy) understanding of the paper (including for the purpose of a review). In particular:

A1. More details, including an algorithm, and more detailed derivations, would be helpful for the lines 161-173 (“Maximum Posterior Inference” for the general case). In particular, what might be an exact algorithm? Can it be reasonably tractable (in terms of computations) and reliable (in terms of convergence)?

A2. More details, in particular an exact algorithm, would be helpful for the lines 204-209 (“Tree Induction” for GRBNs). Can it be reliable (in terms of convergence/accuracy)?

B. The positioning of the RBNs is not clear. In some parts of the paper, it is suggested that RBNs allow to express continuous and discrete variables (e.g. lines 87-89) (and that is the idea proposed in the paper in general), but in some parts it is suggested that RBNs are only for continuous variables (e.g. the position of RBNs in Figure 1; “also is” in line 84 (is “can also be” intended?)).

C. Line 149 refers to the equation (17) as an unnormalised probability distribution but there is a denominator there in the equation; is it (partially) normalised, or is the denominator there for some other purpose? More clarification and maybe more details would help.

D. I am sorry, The sentence in lines 171-173 is not fully clear:

D1. (Just to double check, a minor point.) Is it right to read that the _exponential_ number of network structures is the reason for the “highly non-convex” property? (Just double checking.)

D2. Not clear what is meant by “maybe be highly non-convex” “even for otherwise convex models”.

E. (Just to double check, a minor point.) “the Appendix” in line 192 - is section A.4.2 meant? (A more precise reference might help.)

F. Given that RBNs allow, by definition, different structures (e.g. that a non-terminal variable can have a different number of observed variables), it would be helpful to note/clarify why it is necessary to introduce/use “multi-terminal transitions” and why the “default” feature of RBNs can’t be exploited. (I appreciate that a reason might be that it is easier to advance the limited case of GRBNs with an additional feature because GRBNs allow, with some assumptions, some approximate analytical derivation; or/and because it is a more specific sub-model that might be expected to fit some problems well.)

G. Similarly to the previous point (F), an GRBNs is extended to support categorical (discrete) data but the proposed general case of RBNs can support it “by default”. Some comments on why the GRBNs are extended in that way would be helpful.

H. It would be helpful to clarify how exactly recall and precision are calculated in the experiments.

I. Section 3.1 refers to “Figure 6(a)” (line 280) (but probably some other figure is meant?), and that is also referred to in section 3.2 (line 292) (likely correctly).

J. What is the “RBN estimate” in figure 6(c)? Is it the MAP?

Given the clarity points and the limitations described in another section of the review, although I think the authors have done a lot of work and that is appreciated, sadly I don’t feel comfortable at this point to recommend the submission’s acceptance, I am sorry. There is a chance though that: (a) I did misunderstand some points; (b) some points are obvious and they just were not mentioned (at all/in detail) in the submission; (c) or/and some points are valid and they can be addressed by the camera-ready version submission if the paper is accepted. Given the authors’ rebuttal, to which I am looking forward, there is a chance that the score might be adjusted.

All parts of the review are provided to the best of my understanding and knowledge. Mistakes/misunderstandings in the review are possible. I kindly ask to point me to them, if any, if possible, please.

*

**For the update, please, see my message dated the 2nd of September 2021 (GMT).**


**Time Spent Reviewing:**

I have not tracked time. I generally conduct a review in a few sub-iterations, to be able to reflect (between them) on a submission/material under review for some time.

---

> ### Author Response · Authors · 2021-08-11
> **Point-by-point response**
>
> Many thanks for the detailed feedback, this is much appreciated!
>
> **A1:** For the case of general RBNs with continuous variables, we cannot think of any efficient approach for exact maximum posterior (MAP) inference. As illustrated in Figure 4, even for the fully Gaussian case, the maximum of the marginal posterior (the Gaussian mixture arising from marginalising over all structures) may be unrelated to the maximum of the best structure.
>
> **A2:** Exact tree induction in the sense of exact MAP inference (either jointly with the continuous variables or marginalising them out) is generally intractable (see above). However, in principle, it is possible to find a maximum marginal posterior estimate for the continuous variables via gradient descent (only local convergence guarantees; highly volatile due to interactions with the structural variables; problems with vanishing gradients). This provides the best _distribution_ over trees, from which the single best tree can be efficiently computed (conditional on the continuous variables; i.e. not necessarily the globally best tree) via dynamic programming (one bottom-up/inside pass for evaluating all trees; one top-down/outside pass for selecting the best one).
>
> In our experiments, we instead made use of our closed-form approximation scheme for Gaussian RBNs to select the tree with the maximum (approximate) marginal posterior. As in the final step above, we do one bottom-up/inside pass for evaluating all trees (but here the continuous variables are marginalised out analytically instead of being conditioned on the previously optimised value) and select the best tree in a top-down/outside pass. We will a more detailed explanation of this procedure in the paper.
>
> **B:** RBNs are for both continuous and discrete variable (with the fully discrete case simplifying to PCFGs). In Figure 1, we will clarify that the second "continuous" row also includes the possibility for discrete variables (just like DBN may also include discrete variables) In line 84, "can also be" is indeed the better wording.
>
> **C:** A particularity of RBNs as opposed to normal Bayesian networks is that there is uncertainty about whether a certain variable exists. That is, to specify a variable's marginal, we need to specify its marginal distribution for the case that is _does_ exist but also the probability that it actually does _not_ exist. These two pieces of information are jointly represented by $\widetilde{p}(a_{i:k}|\mathbf{B})$ in equation (17), which is not properly normalised (despite the denominator). The normalisation constant of $\widetilde{p}(a_{i:k}|\mathbf{B})$ corresponds to the probability of $a_{i:k}$ to exist, while the distribution (after being normalised) describes how  $a_{i:k}$ is distributed for the case that is _does_ exist. This is essentially inherited from PCFGs.
>
> **D1:** As soon as we have more than one network structure that we need to marginalise out, we get a mixture. This means that even for the Gaussian case (which for normal Bayesian networks is convex) we get a Gaussian mixture that may be non-convex. The fact that we have an _exponential_ number of network structure just means that any brute force approach (such as iterating over all structures) becomes intractable. It also means that we are likely to get many distinct modes (which we mean by "highly non-convex") unless they happen to all overlap sufficiently to merge into one mode.
>
> **D2:** This refers to the fact that marginalising over structures will likely turn an "otherwise convex model" (such as a Gaussian Bayesian network) into a non-convex model (such as a Gaussian RBN). We will rephrase this sentences to be more clear.
>
> **E:** Yes, line 192 should refer to Appendix A.4.2.
>
> **F:** To keep the overall presentation clear, we mainly focused on the sub-class of RBNs in Chomsky normal form. To be consistent with PCFGs these are defined to only have single-terminal transitions, so that multi-terminal transition come as an extension. But it is perfectly correct that this effectively only comes back to the "default" feature of RBNs, which allow for any distribution over tree structures.
>
> **G:** Here, the point really is that we make use of the Gaussian property to derive our approximation scheme. This fundamentally relies on Gaussians being self-conjugate, so that repeated multiplication and marginalisation can be carried out analytically (interleaved with the approximation steps). The conjugate prior of a categorical distribution is the Dirichlet distribution, which is not conjugate to a Gaussian and thus "breaks" our chain of analytically tractable operations. The corresponding extension of Gaussian RBNs rewrites the Dirichlet as a Gaussian in log-space to make the RBN fully Gaussian and amenable to our approximate inference scheme.
>
> **H:** Precision and recall are computed from the true positive (TP), false positive (FP), and false negative (FN) rates. For the single-tree estimates (baseline model and best-tree estimate from RBNs) we compared the ground-truth and estimated tree node-by-node to count "hits" and "fails". For the marginal node probabilities, we computed the corresponding rates by counting the nodes in the ground-truth tree (TP+FN), summing the marginal probabilities over all nodes in the ground-truth tree (TP), and summing it over all possible nodes (TP+FP). We will add these details to Appendix B.1.
>
> **I:** Thanks for pointing this out. In line 292 the reference is indeed correct, while in line 280 it should in fact be Figure 7 in Appendix B.1.
>
> **J:** It is the tree with maximum marginal likelihood (see our comment on A2 above). We will clarify this.

---

> ### Comment · Reviewer_uEGL · 2021-09-02
> **Update**
>
> Dear Authors, Dear Other Reviewers, Dear Area Chairs, Dear All,
>
> Thanks a lot for the other reviews and the authors' responses.
>
> I think the authors have addressed some of the concerns and they have suggested that some edits could be made to the submission.
>
> My opinion is that although the paper presents some extensive work and it covers quite a few things, acceptance might not be recommended without another review of a modified version, preferably with additional work that could make the paper stronger (in particular, to address the limitations including regarding the empirical evaluation (the current empirical evaluation concerns only continuous variables; also, there is no comparison to existing methods on existing (i.e. independently of the paper) datasets (synthetic or not)); unless the paper is restructured in a different way e.g. by focusing more on the musical application).
>
> I am sorry, I have kept my score at "5: Marginally below the acceptance threshold".
>
> Yours faithfully and sincerely,
> One of the reviewers

---

### Author Response · Authors · 2021-08-11
**Thank you to all reviewers**

We would like to thank all reviewers for their feedback, which is very helpful for further improving the paper. Please see our comments below the respective reviews for a detailed point-by-point response.

---

### Decision · Program_Chairs · 2021-09-28

**Decision:**

Accept (Poster)

**Comment:**

Given the reviewer scores, after responses to the authors rebuttal, it is not possible to accept this paper.  For me the main issue is that the extension of Bayesian networks to PCFGs is an old issue with a large relevant literature. The primary innovation here seems to be the use of continuous random variables.  Continuous random variables in graphical models also has a very large literature. I think the reviewers were not convinced that putting these things together is sufficient for publication at NeurIPS.

**Consistency Experiment:**

NeurIPS has a long history of experimentation. In 2014, NeurIPS ran an experiment in which 10% of submissions were reviewed by two independent committees to quantify the randomness in the review process. This year, we repeated a variant of this experiment to see how the quality of the review process has changed over time.  This paper was part of the experiment and was therefore assigned to two committees (consisting of reviewers, an Area Chair, and a Senior Area Chair) that reached independent decisions.  If both committees made the same recommendation, this recommendation was followed. If a single committee recommended acceptance, the paper was accepted (with the exception of a few cases in which the other committee identified what we considered a fatal flaw, e.g., an error in a key result).

This copy’s committee reached the following decision: **Reject**

The other committee assigned to the paper recommended **Accept (Spotlight)**.  You can find the other set of reviews, along with any follow up discussion with the authors here:
https://openreview.net/forum?id=qdphcA9jEbJ